# Pentacyclic Triterpenoids-Based Ionic Compounds: Synthesis, Study of Structure–Antitumor Activity Relationship, Effects on Mitochondria and Activation of Signaling Pathways of Proliferation, Genome Reparation and Early Apoptosis

**DOI:** 10.3390/cancers15030756

**Published:** 2023-01-26

**Authors:** Lilya U. Dzhemileva, Regina A. Tuktarova, Usein M. Dzhemilev, Vladimir A. D’yakonov

**Affiliations:** N.D. Zelinsky Institute of Organic Chemistry, Russian Academy of Sciences, Leninsky Prospect 47, Moscow 119991, Russia

**Keywords:** ionic compounds, triterpenoids, cytotoxicity, anticancer activity, mitochondrial apoptosis, cell cycle, cell signaling

## Abstract

**Simple Summary:**

The search for new, effective and low-toxicity anticancer drugs is one of the most important tasks of modern medicinal chemistry. Pentacyclic triterpenoids constitute an important class of biologically active compounds with a wide range of biological and pharmacological activities. These compounds are of particular interest due to their antitumor and antiviral properties. Triterpenoids exhibit low toxicity to animals even at high concentrations, but their relatively low potential for biological action, low water solubility and unfavorable absorption and metabolism parameters are a serious obstacle to the use of these substances in clinical practice. In the framework of the presented work, we synthesize a series of ionic compounds based on betulinic, ursolic and oleanolic acids, which could, in our view, improve the solubility of these compounds and their permeability through biological membranes.

**Abstract:**

The present research paper details the synthesis of novel ionic compounds based on triterpene acids (betulinic, oleanolic and ursolic), with these acids acting both as anions and connected through a spacer with various nitrogen-containing compounds (pyridine, piperidine, morpholine, pyrrolidine, triethylamine and dimethylethanolamine) and acting as a cation. Based on the latter, a large number of ionic compounds with various counterions (BF_4_-, SbF_6_-, PF_6_-, CH_3_COO-, C_6_H_5_SO_3_-, *m*-C_6_H_4_(OH)COO- and CH_3_CH(OH)COO-) have been synthesized. We studied the cytotoxicity of the synthesized compounds on the example of various tumor (Jurkat, K562, U937, HL60, A2780) and conditionally normal (HEK293) cell lines. IC50 was determined, and the influence of the structure and nature of the anion and cation on the antitumor activity was specified. Intracellular signaling, apoptosis induction and effects of the most active ionic compounds on the cell cycle and mitochondria have been discussed by applying modern methods of multiparametric enzyme immunoassay and flow cytometry.

## 1. Introduction

Natural pentacyclic triterpenoids of the lupane, ursane and oleanane type, contained in various parts of plants—in the bark, waxy coating of leaves or fruit peel—have an important place among natural compounds of plant origin, as they are considered one of the richest sources of leading structures for the development of new biologically active substances and drugs [1,2,3,4]. Triterpene pentacyclic acids offer a wide variety of biological activities (antitumor, antiviral, antibacterial and antiparasitic), successfully combined with a rather low systemic toxicity [5]. The current drawbacks preventing the promotion of these compounds into clinical practice, such as low biological action potential of native triterpene acids, poor aqueous solubility and insufficient bioavailability from the gastrointestinal tract, stimulate intensive research aimed at modifying them in order to overcome the above disadvantages [6,7,8]. It should be noted that due to the presence of easily transformable functional groups (3-OH, 28-COOH) in the molecules of the above-mentioned natural metabolites, pentacyclic triterpenic acids have a high synthetic potential [4,9]. In this regard, studies aimed at the development of effective approaches and new synthetic methods for the preparation of semisynthetic derivatives of betulinic, ursolic and oleanolic acids, which exhibit high selectivity against biotargets, have acceptable water solubility and the ability to pass through cell membranes, are relevant.

In the framework of the presented work, we suggest the preparation of ionic compounds based on betulinic, ursolic and oleanolic acids, which could, in our view, improve the solubility of these compounds and their permeability through biological membranes.

When setting research objectives, we proceeded from certain assumptions. First, the relative simplicity of the synthesis of ionic compounds makes it possible to obtain sets of substances with both minor and pronounced differences in the structures of the cations and anions in a short time, including the application of pentacyclic triterpene acids, both as an anion and as a cation. Second, there is an increasing trend in the reports on the active exploration of the medical potential of ionic liquids as drugs and drug delivery vehicles [4,10,11,12,13]. Third, numerous studies have demonstrated that the conversion of organic compounds into an ionic form can significantly improve their water solubility, thereby increasing their bioavailability [14,15]. Fourth, the limited information about the mechanisms of action of ionic compounds on living systems, as well as about the mechanisms of action and molecular targets of some well-known medicinal compounds, remains a challenging issue so far.

Consequently, we planned to develop approaches for the synthesis of derivatives of biologically active triterpenoids containing spacers in their structure, ensuring the subsequent introduction of imidazolium, pyridinium, pyrolidinium or piperidinium cations into molecules, for the subsequent synthesis of target ionic liquids on their basis. To study the influence of the nature of the anion on the antitumor activity of molecules, we involved such anions as well as tetrafluoroborate, hexafluorophosphate, acetate, halogen, thiocyanate, etc. In this work, we have synthesized covalently bound triterpenoids (oleanolic, ursolic and betulinic acids) with *N*-methylimidazolium cation, with a different numbers of methylene units separating the triterpenoid and imidazole molecule; ionic compounds based on oleanolic acid covalently bound to various cations (pyridine, piperidine, morpholine, pyrrolidine, triethylamine and dimethylethanolamine); ionic compounds based on imidazole derivatives of oleanolic acid with various anions (BF_4_-, SbF_6_-, PF_6_-, CH_3_COO-, C_6_H_5_SO_3_-, *m*-C_6_H_4_(OH)COO-, CH_3_CH(OH)COO-) and ionic imidazole compounds with triterpenoids (oleanolic, ursolic and betulinic acids) as anions, which made it possible to obtain a fairly large library of ionic compounds from tricyclic terpenoids.

The antitumor activity of the novel compounds was studied with a range of in vitro biological tests. The analysis of the obtained data revealed the correlation between chemical structure of the ionic compound and the cytotoxic reaction caused by it (in particular, signal cascades in cells of various origins), which in turn helps to work out a model for the application of drugs based on ionic liquids in personalized medicine. In each specific case (for each cell line), you can choose the most effective ionic compound, select and synthesize its analogs with minor structural differences and screen the activities of these derivatives. Furthermore, this research project included the study of the mechanisms of antitumor effect of the synthesized ionic compounds by modern methods of flow cytometry and multiparametric analysis using Luminex xMAP technology.

## 2. Materials and Methods

### 2.1. Chemistry

All commercial reagents were purchased from Sigma-Aldrich and Acros organics (Verona, Italy). Betulinic acid was prepared from commercially available betulin by a reported procedure [16]. All commercially available solvents and reagents used were of analytical grade and without further purification. Reactions were monitored by TLC on Sorbfil plates. Column chromatography was carried out on Acrus silica gel (0.060–0.200 mm). Optical rotations were measured on a Perkin–Elmer 341 polarimeter (Waltham, MA, USA). Melting points were recorded on Stuart SMP3. IR spectra were recorded on Bruker VERTEX 70 V using KBr discs over the range of 400–4000 cm^−1^ (Billerica, MA, USA). ^1^H and ^13^C NMR spectra were obtained using a Bruker Ascend 500 spectrometer in CDCl_3_ operating at 500 MHz for ^1^H and 125 MHz for ^13^C and a Bruker AVANCE 400 spectrometer in CDCl_3_ operating at 400 MHz for ^1^H and 100 MHz for ^13^C. Mass spectra of MALDI TOF/TOF positive ions (matrix of sinapic acid) are recorded on a mass spectrometer Bruker AutoflexTM III Smartbeam. Elemental analyses were measured on 1106 Carlo Erba apparatus (Val de Reuil, France). Appendix A show the ^1^H and ^13^C NMR spectra of the synthesized compounds.

### 2.2. Cell Culturing

Human cancer cell line HeLa, HL60, A2780 was obtained from the HPA Culture Collections (Salisbury, UK). Cells (Jurkat, K562, U937, Hek293, Fibroblasts) were purchased from Russian Cell Culture Collection (Institute of Cytology of the Russian Academy of Sciences, Novosibirsk, Russia) and cultured according to standard protocols and sterile technique. The cell lines were shown to be free of viral contamination and mycoplasma. Cells were maintained in RPMI 1640 (Jurkat, K562, U937, HL60) (Gibco, Billings, MT, USA) supplemented with 4 µM glutamine, 10% FBS (Sigma) and 100 units/mL penicillin–streptomycin (Sigma). All types of cells were grown in an atmosphere of 5% CO2 at 37 °C. The cells were subcultured at 2–3 days intervals. Cells were then seeded in 24-well plates at 5 × 10^4^ cells per well and incubated overnight. Jurkat, K562, U937, HL60 cells were subcultured at 2-day intervals with a seeding density of 1 × 10^5^ cells per 24-well plates in RPMI with 10% FBS.

### 2.3. Chemical Experimental Data

#### 2.3.1. General Procedures for Synthesis 6-Hydroxyhexyl (3*β*)-3-(acetyloxy)olean-12-en-28-oate (**2**)

To a solution of (3*β*)-3-(acetyloxy)olean-12-en-28-oic acid **1** (0.49 g, 1 mmol) in anhydrous CH_2_Cl_2_ (20 mL) at 0 °C, oxalyl chloride (1.70 mL, 18.0 mmol) was added. After stirring at room temperature overnight, the mixture was evaporated, and co-evaporated with dry hexane (3 × 10 mL). The residue was dissolved in dry CH_2_Cl_2_ (20 mL), and then DIPEA (0.52 mL, 3.0 mmol) and 1,6-hexanediol (0.24 g, 2.0 mmol) were added at 0 °C. After stirring at rt for 24 h, the solvent was evaporated. The residue was purified by column chromatography to afford the corresponding product **2**.

##### 6-Hydroxyhexyl (3*β*)-3-(acetyloxy)olean-12-en-28-oate (**2**)

Yield: 0.47 g, 78%, white solid, mp 152–154 °C. [α]_D_^22^ + 48.7 (c 0.83, CHCl_3_); IR (KBr) ν_max_ 2943, 2862, 1732, 1619, 1463, 1366, 1246, 1161, 1028, 756, 667, 652 cm^−1^; ^1^H NMR (CHCl_3_, 500 MHz) *δ* 5.28 (1H, m, H-12), 4.49 (1H, br t, *J* = 7.5 Hz, H-3), 4.01 (2H, t, *J* = 6.0 Hz, CH_2_O), 3.64 (2H, t, *J* = 6.5 Hz, CH_2_OH), 2.86 (1H, d, ^2^*J* = 10.0 Hz, H-18), 2.04 (3H, s, CH_3_CO), 2.00–0.81 (22H, m), 1.62 (2H, m, CH_2_), 1.58 (2H, m, CH_2_), 1.39 (4H, m, CH_2_), 1.13 (3H, s, H-27), 0.93 (6H, s, H-25, H-30), 0.90 (3H, s, H-29), 0.87 (3H, s, H-23), 0.85 (3H, s, H-24), 0.73 (3H, s, H-26); ^13^C NMR (CHCl_3_, 125 MHz) *δ* 177.8 (C-28), 171.0 (CH_3_CO), 143.8 (C-13), 122.2 (C-12), 80.9 (C-3), 64.1 (CH_2_O), 62.8 (CH_2_OH), 55.3 (C-5), 47.5 (C-9), 46.7 (C-17), 45.8 (C-19), 41.7 (C-14), 41.3 (C-18), 39.3 (C-8), 38.1 (C-1), 37.7 (C-4), 36.9 (C-10), 33.9 (C-21), 33.1 (C-29), 32.7 (C-7, CH2), 32.5 (C-22), 30.7 (C-20), 28.6 (CH_2_), 28.0 (C-23), 27.6 (C-15), 25.9 (C-27), 25.8 (CH_2_), 25.4 (CH_2_), 23.6 (C-30), 23.5 (C-2), 23.4 (C-11), 22.9 (C-16), 21.3 (CH_3_CO), 18.2 (C-6), 17.0 (C-26), 16.7 (C-24), 15.4 (C-25); anal. calcd for C_38_H_62_O_5_: C, 76.21; H, 10.43; found C, 76.11; H, 10.39. MALDI TOF: *m*/*z* 621.421 ([M]^+^, calcd 621.449).

#### 2.3.2. General Procedures for Synthesis Imidazole Derivatives of Oleanolic Acid (**3**) and (**4**)

To a solution of 6-hydroxyhexyl (3*β*)-3-(acetyloxy)olean-12-en-28-oate **2** (0.40 g, 0.70 mmol) in anhydrous dichloromethane (20 mL) was added the 1-methyl-1*H*-imidazole-2-carboxylic acid (or 1-methyl-1*H*-imidazole-5-carboxylic acid) (0.13 g, 1.10 mmol) followed by *N*-[3-(methylamino)propyl]-*N*’-ethylcarbodiimide hydrochloride (EDC·HCl) (0.21 g, 1.10 mmol) and 4-dimethylaminopyridine (DMAP) (43 mg, 0.35 mmol) under argon. The mixture was stirred at room temperature overnight. The mixture was diluted with H_2_O (10 mL) and the CH_2_Cl_2_ layer was separated, dried over MgSO_4_ and concentrated. The crude product was purified by column chromatography (silica gel) using hexane/EtOAc = 1/2 as the elution solvent to afford imidazole derivatives of oleanolic acid **3** or **4**.

##### 6-{[(1-Methyl-1*H*-imidazol-2-yl)carbonyl]oxy}hexyl (3*β*)-3-(acetyloxy)olean-12-en-28-oate (**3**)

Yield: 0.43 g, 88%, white solid, mp 134–136 °C. [α]_D_^22^ + 30.7 (c 0.57, CHCl_3_); IR (KBr) ν_max_ 2928, 2857, 1717, 1618, 1465, 1422, 1367, 1259, 1159, 1127, 1028, 921, 801, 755, 665 cm^−1^; ^1^H NMR (CHCl_3_, 400 MHz) *δ* 7.15 (1H, m, Imid), 7.04 (1H, m, Imid), 5.28 (1H, m, H-12), 4.49 (1H, t, *J* = 7.5 Hz, H-3), 4.34 (2H, t, *J* = 8.5 Hz, CH2O), 4.02 (3H, s, CH_3_N), 4.01 (2H, m, CH_2_O), 2.87 (1H, d, ^2^*J* = 17.5 Hz, ^3^*J* = 5.0 Hz, H-18), 2.05 (3H, s, CH_3_CO), 2.02–0.81 (22H, m), 1.74 (2H, m, CH_2_), 1.62 (2H, m, CH_2_), 1.45 (4H, m, CH_2_), 1.13 (3H, s, H-27), 0.93 (6H, s, H-25, H-30), 0.90 (3H, s, H-29), 0.87 (3H, s, H-23), 0.85 (3H, s, H-24), 0.73 (3H, s, H-26); ^13^C NMR (CHCl_3_, 100 MHz) *δ* 177.7 (C-28), 170.9 (CH_3_CO), 159.3 (CO_2_), 143.8 (C-13), 136.7 (Imid), 129.5 (Imid), 126.2 (Imid), 122.2 (C-12), 80.9 (C-3), 65.3 (CH_2_O), 64.1 (CH_2_O), 55.3 (C-5), 47.5 (C-9), 46.7 (C-17), 45.8 (C-19), 41.7 (C-14), 41.3 (C-18), 39.3 (C-8), 38.1 (C-1), 37.7 (C-4), 36.9 (C-10), 35.9 (CH_3_N), 33.9 (C-21), 33.1 (C-29), 32.7 (C-7), 32.5 (C-22), 30.7 (C-20), 28.6 (CH_2_), 28.5 (CH_2_), 28.0 (C-23), 27.6 (C-15), 25.8 (CH_2_), 25.6 (C-27, CH_2_), 23.6 (C-30), 23.5 (C-2), 23.4 (C-11), 22.9 (C-16), 21.3 (CH_3_CO), 18.2 (C-6), 17.0 (C-26), 16.7 (C-24), 15.4 (C-25); anal. calcd for C_43_H_66_N_2_O_6_: C, 73.05; H, 9.41; found C, 72.91; H, 9.39. MALDI TOF: *m*/*z* 707.738 ([M + H]^+^, calcd 707.499), 729.725 ([M + Na]^+^, calcd 729.482), 745.709 ([M + K]^+^, calcd 745.456).

##### 6-{[(1-Methyl-1*H*-imidazol-5-yl)carbonyl]oxy}hexyl (3*β*)-3-(acetyloxy)olean-12-en-28-oate (**4**)

Yield: 0.42 g, 85%, white solid, mp 123–125 °C. [α]_D_^20^ + 22.6 (c 0.87, CHCl_3_); IR (KBr) ν_max_ 2946, 2863, 1718, 1541, 1465, 1393, 1365, 1301, 1248, 1226, 1175, 1161, 1128, 985, 922, 756, 656 cm^−1^; ^1^H NMR (CHCl_3_, 500 MHz) *δ* 7.71 (1H, s, Imid), 7.54 (1H, s, Imid), 5.28 (1H, m, H-12), 4.49 (1H, br t, *J* = 7.5 Hz, H-3), 4.26 (2H, t, *J* = 7.0 Hz, CH_2_O), 4.02 (2H, t, *J* = 7.0 Hz, CH_2_O), 3.91 (3H, s, CH_3_N), 2.87 (1H, d, ^2^*J* = 10.0 Hz, H-18), 2.04 (3H, s, CH_3_CO), 2.01–0.81 (22H, m), 1.75 (2H, m, CH_2_), 1.62 (2H, m, CH_2_), 1.45 (4H, m, CH_2_), 1.13 (3H, s, H-27), 0.92 (6H, s, H-25, H-30), 0.89 (3H, s, H-29), 0.86 (3H, s, H-23), 0.85 (3H, s, H-24), 0.73 (3H, s, H-26); ^13^C NMR (CHCl_3_, 125 MHz) *δ* 177.7 (C-28), 170.9 (CH_3_CO), 160.5 (CO_2_), 143.8 (C-13), 142.4 (Imid), 137.5 (Imid), 123.2 (Imid), 122.2 (C-12), 80.9 (C-3), 64.3 (CH_2_O), 64.0 (CH_2_O), 55.3 (C-5), 47.5 (C-9), 46.7 (C-17), 45.8 (C-19), 41.7 (C-14), 41.3 (C-18), 39.3 (C-8), 38.1 (C-1), 37.7 (C-4), 36.9 (C-10), 34.0 (CH_3_N), 33.9 (C-21), 33.1 (C-29), 32.7 (C-7), 32.5 (C-22), 30.7 (C-20), 28.7 (CH_2_), 28.5 (CH_2_), 28.0 (C-23), 27.6 (C-15), 25.8 (CH_2_), 25.6 (C-27, CH_2_), 23.6 (C-30), 23.5 (C-2), 23.4 (C-11), 23.0 (C-16), 21.3 (CH_3_CO), 18.2 (C-6), 17.0 (C-26), 16.7 (C-24), 15.3 (C-25); anal. calcd for C_43_H_66_N_2_O_6_: C, 73.05; H, 9.41; found C, 72.89; H, 9.38. MALDI TOF: *m*/*z* 707.475 ([M + H]^+^, calcd 707.499), 729.763 ([M + Na]^+^, calcd 729.482), 745.438 ([M + K]^+^, calcd 745.456).

#### 2.3.3. General Procedures for Synthesis Bromoalkane Derivatives of Triterpenoids (**6a**–**6d**), (**7a**–**7c**), (**8a**–**8c**)

To a solution of triterpenoid (6.0 mmol) in DMF (10 mL), K_2_CO_3_ (0.84 g, 6.0 mmol) and *α*,*ω*-dibromoalkane (30.0 mmol) were added. After stirring at room temperature for 12 h, the mixture was diluted with H_2_O (50 mL) and extracted with EtOAc (3 × 50 mL). The combined organic layers were washed successively with 1N HCl, H_2_O, saturated aqueous NaHCO_3_ and brine, dried (MgSO_4_), filtered and concentrated. The residue was purified by column chromatography.

##### 2-Bromoethyl (3*β*)-3-hydroxyolean-12-en-28-oate (**6a**)

Yield: 2.03 g, 60%, white solid, mp 169–171 °C (lit. 169–171 °C). [α]_D_^22^ + 54.5 (c 0.87, CHCl_3_); IR (KBr) ν_max_ 2945, 2866, 1726, 1461, 1386, 1364, 1259, 1159, 1124, 1083, 1029, 756, 667, 574 cm^−1^; ^1^H NMR (CDCl_3_, 400 MHz) *δ* 5.32 (1H, m, H-12), 4.41–4.28 (2H, m, CH_2_O), 3.65 (2H, t, *J* = 6.0 Hz, CH_2_Br), 3.22 (1H, m, H-3), 2.89 (1H, dd, ^2^*J* = 19.0 Hz, ^3^*J* = 4.0 Hz, H-18), 2.05–0.72 (22H, m), 1.15 (3H, s, H-27), 1.00 (3H, s, H-23), 0.95 (3H, s, H-30), 0.92 (6H, s, H-25, H-29), 0.79 (3H, s, H-24), 0.76 (3H, s, H-26); ^13^C NMR (CDCl_3_, 100 MHz) *δ* 177.3 (C-28), 143.5 (C-13), 122.6 (C-12), 78.9 (C-3), 63.6 (CH_2_O), 55.2 (C-5), 47.6 (C-9), 46.9 (C-17), 45.8 (C-19), 41.7 (C-14), 41.3 (C-18), 39.4 (C-8), 38.8 (C-4), 38.5 (C-1), 37.0 (C-10), 33.9 (C-21), 33.1 (C-29), 32.8 (C-7), 32.4 (C-22), 30.7 (C-20), 29.0 (CH_2_Br), 28.1 (C-23), 27.7 (C-15), 27.2 (C-2), 25.9 (C-27), 23.6 (C-30), 23.4 (C-11), 22.9 (C-16), 18.3 (C-6), 17.1 (C-26), 15.6 (C-24), 15.3 (C-25); anal. calcd for C_32_H_51_BrO_3_: C, 68.19; H, 9.12; found C, 68.05; H, 9.09. MALDI TOF: *m*/*z* 585.459 ([M + Na]^+^, calcd 585.291).

##### 4-Bromobutyl (3*β*)-3-hydroxyolean-12-en-28-oate (**6b**)

Yield: 2.84 g, 80%, white solid, mp 134–136 °C (lit. 135–137 °C). [α]_D_^26^ + 46.8 (c 0.79, CHCl_3_); IR (KBr) ν_max_ 2945, 2866, 1721, 1462, 1386, 1320, 1259, 1201, 1176, 1161, 1124, 1093, 1032, 826, 756, 655, 562 cm^−1^; ^1^H NMR (CDCl_3_, 500 MHz) *δ* 5.29 (1H, t, *J* = 3.5 Hz, H-12), 4.06 (2H, t, *J* = 6.0 Hz, CH_2_O), 3.44 (2H, t, *J* = 6.0 Hz, CH_2_Br), 3.22 (1H, m, H-3), 2.87 (1H, dd, ^2^*J* = 13.5 Hz, ^3^*J* = 4.0 Hz, H-18), 2.02–0.72 (22H, m), 1.89 (2H, m, CH_2_), 1.63 (2 H, m, CH_2_), 1.15 (3H, s, H-27), 0.99 (3H, s, H-23), 0.94 (3H, s, H-30), 0.92 (6H, s, H-25, H-29), 0.79 (3H, s, H-24), 0.74 (3H, s, H-26); ^13^C NMR (CDCl_3_, 125 MHz) *δ* 177.7 (C-28), 143.8 (C-13), 122.5 (C-12), 78.9 (C-3), 63.2 (CH_2_O), 55.2 (C-5), 47.6 (C-9), 46.7 (C-17), 45.9 (C-19), 41.7 (C-14), 41.3 (C-18), 39.3 (C-8), 38.8 (C-4), 38.4 (C-1), 37.0 (C-10), 33.9 (C-21), 33.1 (C-29, CH_2_Br), 32.7 (C-7), 32.5 (C-22), 30.7 (C-20), 29.5 (CH_2_), 28.1 (C-23), 27.7 (C-15), 27.4 (CH_2_), 27.2 (C-2), 25.9 (C-27), 23.6 (C-30), 23.4 (C-11), 23.0 (C-16), 18.3 (C-6), 17.1 (C-26), 15.6 (C-24), 15.3 (C-25); anal. calcd for C_34_H_55_BrO_3_: C, 69.02; H, 9.37; found C, 68.85; H, 9.35. MALDI TOF: *m*/*z* 613.341 ([M + Na]^+^, calcd 613.323).

##### 6-Bromohexyl (3*β*)-3-hydroxyolean-12-en-28-oate (**6c**)

Yield: 3.60 g, 97%, white solid, mp 72–74 °C (lit. 71–73 °C). [α]_D_^18^ + 37.8 (c 0.73, CHCl_3_); IR (KBr) ν_max_ 2942, 2863, 1719, 1655, 1607, 1525, 1462, 1364, 1262, 1178, 1081, 1010, 756, 654, 562 cm^−1^; ^1^H NMR (CDCl_3_, 500 MHz) *δ* 5.28 (1H, m, H-12), 4.02 (2H, m, CH_2_O), 3.41 (2H, t, *J* = 7.0 Hz, CH_2_Br), 3.21 (1H, m, H-3), 2.87 (1H, dd, ^2^*J* = 13.5 Hz, ^3^*J* = 3.5 Hz, H-18), 2.01–0.72 (22H, m), 1.89 (2H, m, CH_2_), 1.63 (2H, m, CH_2_), 1.48 (2H, m, CH_2_), 1.40 (2H, m, CH_2_), 1.14 (3H, s, H-27), 0.99 (3H, s, H-23), 0.93 (3H, s, H-30), 0.91 (6H, s, H-25, H-29), 0.78 (3H, s, H-24), 0.73 (3H, s, H-26); ^13^C NMR (CDCl_3_, 125 MHz) *δ* 177.7 (C-28), 143.8 (C-13), 122.3 (C-12), 78.9 (C-3), 63.9 (CH_2_O), 55.2 (C-5), 47.6 (C-9), 46.7 (C-17), 45.9 (C-19), 41.7 (C-14), 41.3 (C-18), 39.3 (C-8), 38.7 (C-4), 38.5 (C-1), 37.0 (C-10), 33.9 (C-21), 33.6 (CH_2_Br), 33.1 (C-29), 32.7 (C-7, CH_2_), 32.5 (C-22), 30.7 (C-20), 28.5 (CH_2_), 28.1 (C-23), 27.8 (CH_2_), 27.6 (C-15), 27.2 (C-2), 25.9 (C-27), 25.3 (CH_2_), 23.6 (C-30), 23.4 (C-11), 23.0 (C-16), 18.3 (C-6), 17.0 (C-26), 15.6 (C-24), 15.3 (C-25); anal. calcd for C_36_H_59_BrO_3_: C, 69.54; H, 9.89; found C, 69.39; H, 9.86. MALDI TOF: *m*/*z* 618.501 ([M]^+^, calcd 618.365).

##### 6-Bromohexyl (3*β*)-3-(acetyloxy)olean-12-en-28-oate (**6d**)

Yield: 3.85 g, 97%, white solid, mp 126–128 °C. [α]_D_^21^ + 45.3 (c 0.83, CHCl_3_); IR (KBr) ν_max_ 2945, 2862, 1731, 1619, 1461, 1365, 1245, 1161, 1081, 1029, 728, 669, 560 cm^−1^; ^1^H NMR (CDCl_3_, 500 MHz) *δ* 5.28 (1H, m, H-12), 4.49 (1H, br t, *J* = 7.5 Hz, H-3), 4.02 (2H, t, *J* = 8.0 Hz, CH_2_O), 3.41 (2H, t, *J* = 8.0 Hz, CH_2_Br), 2.87 (1H, dd, ^2^*J* = 17.5 Hz, ^3^*J* = 5.0 Hz, H-18), 2.05 (3H, s, CH_3_CO), 2.02–0.82 (22H, m), 1.89 (2H, m, CH_2_), 1.63 (2H, m, CH_2_), 1.48 (2H, m, CH_2_), 1.40 (2H, m, CH_2_), 1.14 (3H, s, H-27), 0.94 (6H, s, H-25, H-30), 0.91 (3H, s, H-29), 0.87 (3H, s, H-23), 0.86 (3H, s, H-24), 0.74 (3H, s, H-26); ^13^C NMR (CDCl_3_, 125 MHz) *δ* 177.7 (C-28), 170.9 (CH_3_CO), 143.8 (C-13), 122.2 (C-12), 80.9 (C-3), 63.9 (CH_2_O), 55.3 (C-5), 47.5 (C-9), 46.7 (C-17), 45.8 (C-19), 41.7 (C-14), 41.3 (C-18), 39.3 (C-8), 38.1 (C-1), 37.7 (C-4), 36.9 (C-10), 33.9 (C-21), 33.7 (CH_2_Br), 33.1 (C-29), 32.7 (C-7, CH_2_), 32.5 (C-22), 30.7 (C-20), 28.5 (CH_2_), 28.0 (C-23), 27.8 (CH_2_), 27.6 (C-15), 25.8 (C-27), 25.3 (CH_2_), 23.6 (C-30), 23.5 (C-2), 23.4 (C-11), 23.0 (C-16), 21.3 (CH_3_CO), 18.2 (C-6), 17.0 (C-26), 16.7 (C-24), 15.4 (C-25); anal. calcd for C_38_H_61_BrO_3_: C, 68.97; H, 9.29; found C, 68.83; H, 9.26. MALDI TOF: *m*/*z* 660.545 ([M]^+^, calcd 660.375).

##### 2-Bromoethyl (3*β*)-3-hydroxyurs-12-en-28-oate (**7a**)

Yield: 2.06 g, 61%, white solid, mp 68–70 °C. [α]_D_^19^ + 47.4 (c 0.84, CHCl_3_); IR (KBr) ν_max_ 2926, 2870, 1769, 1455, 1386, 1270, 1222, 1180, 1140, 1110, 1029, 996, 756 cm^−1^; ^1^H NMR (CDCl_3_, 400 MHz) *δ* 5.29 (1H,br t, *J* = 3.5 Hz, H-12), 4.33 (2H, t, *J* = 6.0 Hz, CH_2_O), 3.49 (2H, t, *J* = 6.0 Hz, CH_2_Br), 3.22 (1H, m, H-3), 2.26 (1H, d, ^2^*J* = 11.2 Hz, H-18), 2.08–0.72 (22H, m), 1.10 (3H, s, H-27), 1.00 (3H, s, H-23), 0.95 (3H, d, *J* = 6.5 Hz, H-30), 0.93 (3H, s, H-25), 0.87 (3H, d, *J* = 6.0 Hz, H-29), 0.79 (3H, s, H-24), 0.76 (3H, s, H-26); ^13^C NMR (CDCl_3_, 100 MHz) *δ* 177.1 (C-28), 137.9 (C-13), 125.9 (C-12), 79.0 (C-3), 63.7 (CH_2_O), 55.2 (C-5), 52.8 (C-18), 48.3 (C-17), 47.6 (C-9), 42.1 (C-14), 39.6 (C-8), 39.1 (C-19), 38.8 (C-20), 38.7 (C-4), 38.6 (C-1), 36.9 (C-10), 36.7 (C-22), 33.0 (C-7), 30.7 (C-21), 29.0 (CH_2_Br), 28.2 (C-23), 28.0 (C-15), 27.2 (C-2), 24.2 (C-16), 23.5 (C-27), 23.3 (C-11), 21.2 (C-30), 18.3 (C-6), 17.2 (C-29), 16.9 (C-26), 15.6 (C-24), 15.5 (C-25); anal. calcd for C_32_H_51_BrO_3_: C, 68.19; H, 9.12; found C, 68.01; H, 9.08. MALDI TOF: *m*/*z* 585.329 ([M + Na]^+^, calcd 585.291).

##### 4-Bromobutyl (3*β*)-3-hydroxyurs-12-en-28-oate (**7b**)

Yield: 2.84 g, 80%, white solid, mp 60–62 °C. [α]_D_^19^ + 43.6 (c 0.92, CHCl_3_); IR (KBr) ν_max_ 2926, 2871, 1719, 1455, 1386, 1270, 1229, 1198, 1167, 1141, 1111, 1030, 996, 756 cm^−1^; ^1^H NMR (CDCl_3_, 500 MHz) *δ* 5.26 (1H, t, *J* = 3.6 Hz, H-12), 4.09–3.97 (2H, m, CH_2_O), 3.44 (2H, t, *J* = 6.5 Hz, CH_2_Br), 3.22 (1H, m, H-3), 2.24 (1H, d, ^2^*J* = 11.0 Hz, H-18), 2.06–0.72 (22H, m), 1.95 (2H, m, CH_2_), 1.79 (2H, m, CH_2_), 1.10 (3H, s, H-27), 1.00 (3H, s, H-23), 0.96 (3H, d, *J* = 6.0 Hz, H-30), 0.94 (3H, s, H-25), 0.88 (3H, d, *J* = 6.0 Hz, H-29), 0.79 (3H, s, H-24), 0.78 (3H, s, H-26); ^13^C NMR (CDCl_3_, 125 MHz) *δ* 177.5 (C-28), 138.2 (C-13), 125.6 (C-12), 78.9 (C-3), 63.2 (CH_2_O), 55.2 (C-5), 52.9 (C-18), 48.1 (C-17), 47.5 (C-9), 42.1 (C-14), 39.6 (C-8), 39.1 (C-19), 38.9 (C-20), 38.8 (C-4), 38.6 (C-1), 36.9 (C-10), 36.8 (C-22), 33.1 (CH_2_Br), 33.0 (C-7), 30.7 (C-21), 29.5 (CH_2_), 28.2 (C-23), 27.9 (C-15), 27.2 (CH_2_), 27.2 (C-2), 24.2 (C-16), 23.6 (C-27), 23.3 (C-11), 21.2 (C-30), 18.3 (C-6), 17.2 (C-29), 17.0 (C-26), 15.6 (C-24), 15.5 (C-25); anal. calcd for C_34_H_55_BrO_3_: C, 69.02; H, 9.37; found C, 68.89; H, 9.34. MALDI TOF: *m*/*z* 613.429 ([M + Na]^+^, calcd 613.323).

##### 6-Bromohexyl (3*β*)-3-hydroxyurs-12-en-28-oate (**7c**)

Yield: 3.30 g, 89%, white solid, mp 66–68 °C. [α]_D_^21^ + 40.0 (c 0.84, CHCl_3_); IR (KBr) ν_max_ 2927, 2868, 1719, 1456, 1386, 1269, 1231, 1199, 1168, 1142, 1112, 1043, 996, 756 cm^−1^; ^1^H NMR (CDCl_3_, 500 MHz) *δ* 5.25 (1H, t, *J* = 3.5 Hz, H-12), 4.00 (2H, m, CH_2_O), 3.42 (2H, t, *J* = 7.0 Hz, CH_2_Br), 3.22 (1H, m, H-3), 2.24 (1H, d, ^2^*J* = 11.0 Hz, H-18), 2.06–0.72 (22H, m), 1.88 (2H, m, CH_2_), 1.63 (2H, m, CH_2_), 1.47 (2H, m, CH_2_), 1.40 (2H, m, CH_2_), 1.09 (3H, s, H-27), 1.00 (3H, s, H-23), 0.97 (3H, d, *J* = 6.0 Hz, H-30), 0.93 (3H, s, H-25), 0.88 (3H, d, *J* = 6.5 Hz, H-29), 0.79 (3H, s, H-24), 0.77 (3H, s, H-26); ^13^C NMR (CDCl_3_, 125 MHz) *δ* 177.6 (C-28), 138.2 (C-13), 125.5 (C-12), 79.0 (C-3), 63.9 (CH_2_O), 55.2 (C-5), 52.9 (C-18), 48.1 (C-17), 47.6 (C-9), 42.1 (C-14), 39.6 (C-8), 39.1 (C-19), 38.9 (C-20), 38.8 (C-4), 38.6 (C-1), 36.9 (C-10), 36.8 (C-22), 33.7 (CH_2_Br), 33.1 (C-7), 32.7 (CH_2_), 30.7 (C-21), 28.4 (CH_2_), 28.2 (C-23), 27.9 (C-15), 27.8 (CH_2_), 27.2 (C-2), 25.3 (CH_2_), 24.2 (C-16), 23.6 (C-27), 23.3 (C-11), 21.2 (C-30), 18.3 (C-6), 17.2 (C-29), 17.0 (C-26), 15.6 (C-24), 15.5 (C-25); anal. calcd for C_36_H_59_BrO_3_: C, 69.77; H, 9.60; found C, 69.59; H, 9.54. MALDI TOF: *m*/*z* 641.456 ([M + Na]^+^, calcd 641.365).

##### 2-Bromoethyl (3*β*)-3-hydroxylup-20(29)-en-28-oate (**8a**)

Yield: 2.09 g, 62%, white solid, mp 176–178 °C. [α]_D_^21^ + 3.9 (c 0.54, CHCl_3_); IR (KBr) ν_max_ 2942, 2868, 1724, 1454, 1377, 1270, 1152, 1129, 1044, 1032, 946, 885, 756, 664 cm^−1^; ^1^H NMR (CDCl_3_, 500 MHz) *δ* 4.75 (1H, br s, H-29), 4.62 (1H, br s, H-29), 4.42 (2H, m, CH_2_O), 3.55 (2H, t, *J* = 6.0 Hz, CH_2_Br), 3.19 (1H, m, H-3), 3.03 (1H, m, H-19), 2.32–0.68 (24H, m), 1.70 (3H, s, H-30), 0.98 (6H, s, H-23, H-27), 0.94 (3H, s, H-26), 0.83 (3H, s, H-25), 0.77 (3H, s, H-24); ^13^C NMR (CDCl_3_, 125 MHz) *δ* 175.7 (C-28), 150.4 (C-20), 109.7 (C-29), 78.9 (C-3), 63.3 (CH_2_O), 56.7 (C-17), 55.4 (C-5), 50.6 (C-9), 49.4 (C-18), 46.9 (C-19), 42.4 (C-14), 40.7 (C-8), 38.9 (C-4), 38.7 (C-1), 38.3 (C-13), 37.2 (C-22), 37.0 (C-10), 34.3 (C-7), 32.1 (C-16), 30.6 (C-21), 29.7 (C-15), 29.2 (CH_2_Br), 27.9 (C-23), 27.4 (C-2), 25.5 (C-12), 20.9 (C-11), 19.4 (C-30), 18.3 (C-6), 16.1 (C-25), 16.0 (C-26), 15.4 (C-24), 14.7 (C-27); anal. calcd for C_32_H_51_BrO_3_: C, 68.19; H, 9.12; found C, 68.01; H, 9.08. MALDI TOF: *m*/*z* 585.449 ([M + Na]^+^, calcd 585.292).

##### 2-Bromobutyl (3*β*)-3-hydroxylup-20(29)-en-28-oate (**8b**)

Yield: 2.98 g, 84%, white solid, mp 70–72 °C. [α]_D_^21^ + 3.5 (c 0.69, CHCl_3_); IR (KBr) ν_max_ 2944, 2869, 1723, 1452, 1376, 1250, 1154, 1133, 1043, 983, 884, 756, 648 cm^−1^; ^1^H NMR (CDCl_3_, 400 MHz) *δ* 4.75 (1H, br s, H-29), 4.62 (1H, br s, H-29), 4.13 (2H, m, CH_2_O), 3.46 (2H, t, *J* = 6.8 Hz, CH_2_Br), 3.19 (1H, m, H-3), 3.01 (1H, m, H-19), 2.28–0.67 (24H, m), 1.95 (2H, m, CH_2_), 1.79 (2H, m, CH_2_), 1.70 (3H, s, H-30), 0.98 (6H, s, H-23, H-27), 0.93 (3H, s, H-26), 0.84 (3H, s, H-25), 0.77 (3H, s, H-24); ^13^C NMR (CDCl_3_, 100 MHz) *δ* 176.1 (C-28), 150.5 (C-20), 109.6 (C-29), 78.9 (C-3), 62.8 (CH_2_O), 56.6 (C-17), 55.4 (C-5), 50.6 (C-9), 49.4 (C-18), 47.0 (C-19), 42.4 (C-14), 40.7 (C-8), 38.9 (C-4), 38.7 (C-1), 38.3 (C-13), 37.2 (C-22), 37.1 (C-10), 34.3 (C-7), 33.0 (CH_2_Br), 32.2 (C-16), 30.6 (C-21), 29.7 (C-15), 29.5 (CH_2_), 27.9 (C-23), 27.5 (CH_2_), 27.4 (C-2), 25.6 (C-12), 20.9 (C-11), 19.4 (C-30), 18.3 (C-6), 16.1 (C-25), 16.0 (C-26), 15.4 (C-24), 14.7 (C-27); anal. calcd for C_34_H_55_BrO_3_: C, 69.02; H, 9.37; found C, 68.81; H, 9.33. MALDI TOF: *m*/*z* 629.316 ([M + K]^+^, calcd 629.297).

##### 2-Bromohexyl (3*β*)-3-hydroxylup-20(29)-en-28-oate (**8c**)

Yield: 3.45 g, 93%, white solid, mp 68–70 °C. [α]_D_^21^ + 6.3 (c 0.79, CHCl_3_); IR (KBr) ν_max_ 2940, 2867, 1721, 1454, 1377, 1268, 1152, 1043, 983, 884, 758, 664 cm^−1^; ^1^H NMR (CDCl_3_, 500 MHz) *δ* 4.74 (1H, br s, H-29), 4.61 (1H, br s, H-29), 4.14–4.04 (2H, m, CH_2_O), 3.42 (2H, t, *J* = 7.0 Hz, CH_2_Br), 3.19 (1H, m, H-3), 3.02 (1H, m, H-19), 2.28–0.67 (24H, m), 1.90 (2H, m, CH_2_), 1.66 (2H, m, CH_2_), 1.50 (2H, m, CH_2_), 1.42 (2H, m, CH_2_), 1.69 (3H, s, H-30), 0.98 (6H, s, H-23, H-27), 0.93 (3H, s, H-26), 0.83 (3H, s, H-25), 0.77 (3H, s, H-24); ^13^C NMR (CDCl_3_, 125 MHz) *δ* 176.2 (C-28), 150.6 (C-20), 109.6 (C-29), 78.9 (C-3), 63.7 (CH_2_O), 56.5 (C-17), 55.4 (C-5), 50.6 (C-9), 49.4 (C-18), 47.0 (C-19), 42.4 (C-14), 40.7 (C-8), 38.9 (C-4), 38.7 (C-1), 38.3 (C-13), 37.2 (C-22), 37.1 (C-10), 34.3 (C-7), 33.6 (CH_2_Br), 32.7 (CH_2_), 32.2 (C-16), 30.7 (C-21), 29.6 (C-15), 28.6 (CH_2_), 27.9 (C-23), 27.8 (CH_2_), 27.4 (C-2), 25.6 (C-12), 25.3 (CH_2_), 20.9 (C-11), 19.4 (C-30), 18.3 (C-6), 16.1 (C-25), 16.0 (C-26), 15.4 (C-24), 14.7 (C-27); anal. calcd for C_36_H_59_BrO_3_: C, 68.19; H, 9.12; found C, 68.01; H, 9.08. MALDI TOF: *m*/*z* 657.561 ([M + K]^+^, calcd 657.328).

#### 2.3.4. General Synthetic Procedure for Ionic Compounds (ICs) with Triterpenoids Covalently Linked to the *N*-Methylimidazole Cation (**9a**–**9d**), (**10a**–**10c**), (**11a**–**11c**)

To a solution of bromoalkane derivative of triterpenoid **6a**–**6d** (or **7a**–**7c**, **8a**–**8c**) (1 mmol) in dry MeCN was added *N*-methylimidazole (1 mmol, 82 mg) under argon. After stirring at 80 °C for 4–5 days, the solvent was evaporated. The residue was purified by column chromatography to obtain the corresponding product **9a**–**9d** (or **10a**–**10c**, **11a**–**11c**) using hexane/EtOAc (1/1) as the elution solvent; then, CHCl_3_/MeOH (10/1) was used.

##### 1-(2-{[(3*β*)-3-Hydroxyolean-12-en-28-oyl]oxy}ethyl)-3-methylimidazolium Bromide (**9a**) [EMIM-O-Olean][Br]

Yield: 0.49 g, 77%, white solid, mp 254–256 °C. [α]_D_^22^ + 40.3 (c 0.74, MeOH); IR (KBr) ν_max_ 2925, 2855, 1718, 1637, 1578, 1461, 1377, 1318, 1253, 1172, 1044, 751, 722, 654, 619 cm^−1^; ^1^H NMR (DMSO-d_6_, 400 MHz) *δ* 9.29 (1H, s, Imid), 7.82 (1H, m, Imid), 7.79 (1H, m, Imid), 5.08 (1H, m, H-12), 4.50 (2H, m, CH_2_N), 4.30 (2H, m, CH_2_O), 3.87 (3H, s, NCH_3_), 2.99 (1H, m, H-3), 2.65 (1H, d, ^2^*J* = 10.0 Hz, H-18), 1.95–0.62 (22H, m), 1.04 (3H, s, H-27), 0.88 (3H, s, H-23), 0.86 (3H, s, H-29), 0.85 (3H, s, H-30), 0.82 (3H, s, H-25), 0.67 (3H, s, H-24), 0.46 (3H, s, H-26); ^13^C NMR (DMSO-d_6_, 100 MHz) *δ* 176.7 (C-28), 143.6 (C-13), 137.4 (Imid), 124.1 (Imid), 123.1 (Imid), 122.4 (C-12), 77.3 (C-3), 62.8 (CH_2_O), 55.2 (C-5), 48.3 (CH_2_N), 47.4 (C-9), 46.6 (C-17), 45.7 (C-19), 41.5 (C-14), 41.2 (C-18), 39.1 (C-8), 38.8 (C-4), 38.5 (C-1), 36.9 (C-10), 36.3 (CH_3_N), 33.5 (C-21), 33.3 (C-29), 32.7 (C-7), 32.3 (C-22), 30.7 (C-20), 28.7 (C-23), 27.4 (C-15), 27.3 (C-2), 26.0 (C-27), 23.8 (C-30), 23.3 (C-11), 22.8 (C-16), 18.4 (C-6), 16.9 (C-26), 16.5 (C-24), 15.5 (C-25); anal. calcd for C_36_H_57_BrN_2_O_3_: C, 66.96; H, 8.90; found C, 66.86; H, 8.88. MALDI TOF: *m*/*z* 565.452 (calculated for cation C_36_H_57_N_2_O_3_ [M]^+^, calcd 565.437).

##### 1-(4-{[(3*β*)-3-Hydroxyolean-12-en-28-oyl]oxy}butyl)-3-methylimidazolium bromide (**9b**) [BMIM-O-Olean][Br]

Yield: 0.57 g, 85%, white solid, mp 148–150 °C. [α]_D_^24^ + 44.0 (c 0.84, MeOH); IR (KBr) ν_max_ 2946, 2862, 1711, 1637, 1574, 1462, 1385, 1318, 1261, 1211, 1166, 1045, 756, 662, 624 cm^−1^; ^1^H NMR (DMSO-d_6_, 500 MHz) *δ* 9.17 (1H, s, Imid), 7.77 (1H, m, Imid), 7.73 (1H, m, Imid), 5.16 (1H, m, H-12), 4.36 (1H, m, OH), 4.21 (2H, t, *J* = 7.0 Hz, CH_2_N), 3.96 (2H, t, *J* = 6.0 Hz, CH_2_O), 3.85 (3H, s, NCH_3_), 2.99 (1H, m, H-3), 2.76 (1H, d, ^2^*J* = 10.0 Hz, H-18), 1.98–0.62 (22H, m), 1.83 (2H, m, CH_2_), 1.52 (2H, m, CH_2_), 1.08 (3H, s, H-27), 0.88 (3H, s, H-23), 0.87 (6H, s, H-29, H-30), 0.82 (3H, s, H-25), 0.67 (3H, s, H-24), 0.61 (3H, s, H-26); ^13^C NMR (DMSO-d_6_, 125 MHz) *δ* 176.7 (C-28), 143.6 (C-13), 137.4 (CH=), 124.1 (CH=), 123.1 (CH=), 122.4 (C-12), 77.3 (C-3), 62.8 (CH_2_O), 55.2 (C-5), 48.3 (CH_2_N), 47.4 (C-9), 46.6 (C-17), 45.7 (C-19), 41.5 (C-14), 41.2 (C-18), 39.1 (C-8), 38.8 (C-4), 38.5 (C-1), 36.9 (C-10), 36.3 (CH_3_N), 33.5 (C-21), 33.3 (C-29), 32.7 (C-7), 32.3 (C-22), 30.7 (C-20), 28.7 (C-23), 27.4 (C-15), 27.3 (C-2), 26.0 (C-27), 23.8 (C-30), 23.3 (C-11), 22.8 (C-16), 18.4 (C-6), 16.9 (C-26), 16.5 (C-24), 15.5 (C-25); anal. calcd for C_38_H_61_BrN_2_O_3_: C, 67.74; H, 9.12; found C, 67.68; H, 9.09. MALDI TOF: *m*/*z* 593.482 (calculated for cation C_38_H_61_N_2_O_3_ [M]^+^, calcd 593.468).

##### 1-(6-{[(3*β*)-3-Hydroxyolean-12-en-28-oyl]oxy}hexyl)-3-methylimidazolium bromide (**9c**) [HMIM-O-Olean][Br]

Yield: 0.66 g, 94%, white solid, mp 130–132 °C. [α]_D_^25^ + 95.5 (c 0.71, MeOH); IR (KBr) ν_max_ 2931, 2864, 1715, 1622, 1573, 1463, 1386, 1364, 1262, 1238, 1202, 1164, 1032, 756, 664, 623 cm^−1^; ^1^H NMR (DMSO-d_6_, 500 MHz) *δ* 9.16 (1H, s, Imid), 7.77 (1H, m, Imid), 7.71 (1H, m, Imid), 5.17 (1H, m, H-12), 4.36 (1H,br d, *J* = 3.6 Hz, OH), 4.16 (2H, t, *J* = 7.0 Hz, CH_2_N), 3.93 (2H, t, *J* = 6.5 Hz, CH_2_O), 3.85 (3H, s, NCH_3_), 2.99 (1H, m, H-3), 2.77 (1H, d, ^2^*J* = 10.0 Hz, H-18), 1.98–0.62 (22H, m), 1.77 (2H, m, CH_2_), 1.55 (2H, m, CH_2_), 1.35 (2H, m, CH_2_), 1.25 (2H, m, CH_2_), 1.08 (3H, s, H-27), 0.88 (3H, s, H-23), 0.87 (6H, s, H-29, H-30), 0.82 (3H, s, H-25), 0.67 (3H, s, H-24), 0.61 (3H, s, H-26); ^13^C NMR (DMSO-d_6_, 125 MHz) *δ* 177.1 (C-28), 143.9 (C-13), 136.9 (Imid), 124.0 (Imid), 122.7 (Imid), 122.3 (C-12), 77.3 (C-3), 64.0 (CH_2_O), 55.2 (C-5), 49.2 (CH_2_N), 47.5 (C-9), 46.5 (C-17), 45.8 (C-19), 41.7 (C-14), 41.4 (C-18), 39.3 (C-8), 38.8 (C-4), 38.5 (C-1), 36.9 (C-10), 36.2 (CH_3_N), 33.6 (C-21), 33.2 (C-29), 32.8 (C-7), 32.6 (C-22), 30.8 (C-20), 29.8 (CH_2_), 28.7 (C-23), 28.3 (CH_2_), 27.5 (C-15), 27.4 (C-2), 26.0 (C-27), 25.6 (CH_2_), 25.4 (CH_2_), 23.8 (C-30), 23.4 (C-11), 23.0 (C-16), 18.4 (C-6), 17.2 (C-26), 16.5 (C-24), 15.5 (C-25); anal. calcd for C_40_H_65_BrN_2_O_3_: C, 68.45; H, 9.33; found C, 68.38; H, 9.29. MALDI TOF: *m*/*z* 621.487 (calculated for cation C_40_H_65_N_2_O_3_ [M]^+^, calcd 621.499).

##### 1-(6-{[(3*β*)-3-Acetyloxyolean-12-en-28-oyl]oxy}hexyl)-3-methylimidazolium bromide (**9d**) [HMIM-O-AcOlean][Br]

Yield: 0.71 g, 95%, white solid, mp 151–153 °C. [α]_D_^18^ + 32.0 (c 0.71, MeOH); IR (KBr) ν_max_ 2945, 2863, 1729, 1640, 1573, 1463, 1385, 1366, 1247, 1238, 1201, 1173, 1028, 754, 665, 621 cm^−1^; ^1^H NMR (DMSO-d_6_, 400 MHz) *δ* 9.19 (1H, s, Imid), 7.78 (1H, m, Imid), 7.73 (1H, m, Imid), 5.17 (1H, m, H-12), 4.38 (1H, m, H-3), 4.17 (2H, t, *J* = 7.2 Hz, CH2N), 3.98 (2H, t, *J* = 6.4 Hz, CH_2_O), 3.86 (3H, s, NCH_3_), 2.76 (1H, d, ^2^*J* = 10.5 Hz, H-18), 1.99 (3H, s, CH_3_), 1.99–0.80 (22H, m), 1.77 (2H, m, CH_2_), 1.55 (2H, m, CH_2_), 1.35 (2H, m, CH_2_), 1.25 (2H, m, CH_2_), 1.11 (3H, s, H-27), 0.86 (9H, s, H-25, H-29, H-30), 0.80 (3H, s, H-23, H-24), 0.65 (3H, s, H-26); ^13^C NMR (DMSO-d_6_, 100 MHz) *δ* 177.0 (C-28), 170.6 (CH_3_CO), 143.9 (C-13), 136.9 (Imid), 124.0 (Imid), 122.7 (Imid), 122.2 (C-12), 80.3 (C-3), 64.0 (CH_2_O), 54.9 (C-5), 49.2 (CH_2_N), 47.3 (C-9), 46.5 (C-17), 45.8 (C-19), 41.7 (C-14), 41.4 (C-18), 39.3 (C-8), 38.0 (C-1), 37.7 (C-4), 36.9 (C-10), 36.2 (CH_3_N), 33.6 (C-21), 33.2 (C-29), 32.7 (C-7), 32.5 (C-22), 30.8 (C-20), 29.8 (CH_2_), 28.3 (C-23, CH_2_), 27.5 (C-15), 25.9 (C-27), 25.8 (CH_2_), 25.4 (CH_2_), 23.8 (C-30), 23.6 (C-2), 23.4 (C-11), 22.9 (C-16), 21.3 (CH_3_CO), 18.2 (C-6), 17.1 (C-24, C-26), 15.5 (C-25); anal. calcd for C_42_H_67_BrN_2_O_4_: C, 67.81; H, 9.08; found C, 67.71; H, 9.06. MALDI TOF: *m*/*z* 663.407 (calculated for cation C_42_H_67_N_2_O_4_ [M]^+^, calcd 663.510).

##### 1-(2-{[(3*β*)-3-Hydroxyurs-12-en-28-oyl]oxy}ethyl)-3-methylimidazolium bromide (**10a**) [EMIM-O-Urs][Br]

Yield: 0.49 g, 76%, white solid, mp 260–262 °C. [α]_D_^19^ + 2.9 (c 0.87, MeOH); IR (KBr) ν_max_ 2941, 2868, 1728, 1576, 1452, 1385, 1165, 1133, 1046, 879, 750, 621 cm^−1^; ^1^H NMR (DMSO-d_6_, 400 MHz) *δ* 9.39 (1H, s, Imid), 7.87 (1H, m, Imid), 7.83 (1H, m, Imid), 4.99 (1H, m, H-12), 4.51 (2H, m, CH_2_N), 4.25 (2H, m, CH_2_O), 3.89 (3H, s, CH_3_N), 3.00 (1H, m, H-3), 2.05 (1H, d, ^2^*J* = 11.2 Hz, H-18), 2.00–0.63 (22H, m), 0.99 (3H, s, H-27), 0.90 (3H, d, *J* = 6.5 Hz, H-30), 0.89 (3H, s, H-23), 0.83 (3H, s, H-25), 0.79 (3H, d, *J* = 6.4 Hz, H-29), 0.63 (3H, s, H-24), 0.48 (3H, s, H-26); ^13^C NMR (DMSO-d_6_, 100 MHz) *δ* 176.4 (C-28), 138.2 (Imid), 137.6 (C-13), 125.4 (C-12), 124.2 (Imid), 123.2 (Imid), 77.2 (C-3), 62.9 (CH_2_O), 55.2 (C-5), 52.7 (C-18), 48.2 (CH_2_N), 47.9 (C-17), 47.3 (C-9), 41.9 (C-14), 39.4 (C-8), 38.8 (C-4, C-19, C-20), 38.7 (C-1), 36.9 (C-10), 36.5 (C-22), 36.3 (CH_3_N), 32.9 (C-7), 30.4 (C-21), 28.7 (C-23), 27.7 (C-15), 27.4 (C-2), 24.0 (C-16), 23.7 (C-27), 23.2 (C-11), 21.4 (C-30), 18.4 (C-6), 17.3 (C-29), 16.9 (C-26), 16.6 (C-24), 15.7 (C-25); anal. calcd for C_36_H_57_BrN_2_O_3_: C, 66.96; H, 8.90; found C, 66.79; H, 8.81. MALDI TOF: *m*/*z* 565.523 (calculated for cation C_36_H_57_N_2_O_3_ [M + Na]^+^, calcd 565.437).

##### 1-(2-{[(3*β*)-3-Hydroxyurs-12-en-28-oyl]oxy}butyl)-3-methylimidazolium bromide (**10b**) [BMIM-O-Urs][Br]

Yield: 0.57 g, 84%, white solid, mp 114–116 °C. [α]_D_^19^ + 40.9 (c 0.82, MeOH); IR (KBr) ν_max_ 2926, 2871, 1720, 1573, 1454, 1385, 1230, 1168, 1143, 1090, 1046, 880, 757, 622 cm^−1^; ^1^H NMR (DMSO-d_6_, 400 MHz) *δ* 9.29 (1H, s, Imid), 7.83 (1H, m, Imid), 7.78 (1H, m, Imid), 5.13 (1H, m, H-12), 4.31 (1H, m, OH), 4.22 (2H, m, CH_2_N), 3.93 (2H, m, CH_2_O), 3.87 (3H, s, CH_3_N), 2.99 (1H, m, H-3), 2.13 (1H, d, ^2^*J* = 10.0 Hz, H-18), 2.02–0.63 (22H, m), 1.83 (2H, m, CH_2_), 1.52 (2H, m, CH_2_),1.04 (3H, s, H-27), 0.91 (3H, d, *J* = 6.5 Hz, H-30), 0.89 (3H, s, H-23), 0.83 (3H, s, H-25), 0.82 (3H, d, *J* = 6.4 Hz, H-29), 0.66 (3H, s, H-24), 0.64 (3H, s, H-26); ^13^C NMR (DMSO-d_6_, 100 MHz) *δ* 176.8 (C-28), 138.4 (Imid), 137.1 (C-13), 125.4 (C-12), 124.1 (Imid), 122.7 (Imid), 77.2 (C-3), 63.5 (CH_2_O), 55.5 (C-5), 52.9 (C-18), 48.8 (CH_2_N), 47.9 (C-17), 47.4 (C-9), 42.0 (C-14), 39.5 (C-8), 38.8 (C-4, C-19, C-20), 38.7 (C-1), 36.9 (C-10), 36.7 (C-22), 36.3 (CH_3_N), 33.1 (C-7), 30.5 (C-21), 28.7 (C-23), 27.9 (C-15), 27.4 (C-2), 26.8 (CH_2_), 25.3 (CH_2_), 24.2 (C-16), 23.7 (C-27), 23.3 (C-11), 21.4 (C-30), 18.4 (C-6), 17.4 (C-29), 17.2 (C-26), 16.5 (C-24), 15.7 (C-25); anal. calcd for C_38_H_61_BrN_2_O_3_: C, 67.74; H, 9.12; found C, 67.59; H, 9.07. MALDI TOF: *m*/*z* 593.448 (calculated for cation C_38_H_61_N_2_O_3_ [M + Na]^+^, calcd 593.468).

##### 1-(2-{[(3*β*)-3-Hydroxyurs-12-en-28-oyl]oxy}hexyl)-3-methylimidazolium bromide (**10c**) [HMIM-O-Urs][Br]

Yield: 0.65 g, 93%, white solid, mp 118–120 °C. [α]_D_^21^ + 33.0 (c 0.86, MeOH); IR (KBr) ν_max_ 2923, 2854, 1715, 1569, 1461, 1377, 1270, 1231, 1166, 1044, 1029, 995, 740, 621 cm^−1^; ^1^H NMR (DMSO-d_6_, 400 MHz) *δ* 9.26 (1H, s, Imid), 7.82 (1H, m, Imid), 7.76 (1H, m, Imid), 5.14 (1H, m, H-12), 4.31 (1H, br d, *J* = 4.8 Hz, OH), 4.18 (2H, t, *J* = 6.8 Hz, CH_2_N), 3.92 (2H, m, CH_2_O), 3.87 (3H, s, CH_3_N), 2.99 (1H, m, H-3), 2.13 (1H, d, ^2^*J* = 11.2 Hz, H-18), 2.02–0.63 (22H, m), 1.78 (2H, m, CH_2_), 1.55 (2H, m, CH_2_), 1.35 (2H, m, CH_2_), 1.25 (2H, m, CH_2_), 1.05 (3H, s, H-27), 0.91 (3H, d, *J* = 6.5 Hz, H-30), 0.89 (3H, s, H-23), 0.84 (3H, s, H-25), 0.82 (3H, d, *J* = 6.0 Hz, H-29), 0.67 (6H, s, H-24, H-26); ^13^C NMR (DMSO-d_6_, 100 MHz) *δ* 176.8 (C-28), 138.4 (Imid), 137.0 (C-13), 125.4 (C-12), 124.0 (Imid), 122.7 (Imid), 77.2 (C-3), 63.9 (CH_2_O), 55.2 (C-5), 52.9 (C-18), 49.1 (CH_2_N), 47.8 (C-17), 47.4 (C-9), 42.0 (C-14), 39.4 (C-8), 38.8 (C-4, C-19, C-20), 38.7 (C-1), 36.9 (C-10), 36.8 (C-22), 36.3 (CH_3_N), 33.2 (C-7), 30.5 (C-21), 29.8 (CH_2_), 28.7 (C-23), 28.3 (CH_2_), 27.9 (C-15), 27.4 (C-2), 25.6 (CH_2_), 25.4 (CH_2_), 24.2 (C-16), 23.7 (C-27), 23.3 (C-11), 21.4 (C-30), 18.4 (C-6), 17.4 (C-29), 17.2 (C-26), 16.6 (C-24), 15.6 (C-25); anal. calcd for C_40_H_65_BrN_2_O_3_: C, 68.45; H, 9.33; found C, 68.29; H, 9.30. MALDI TOF: *m*/*z* 621.348 (calculated for cation C_38_H_61_N_2_O_3_ [M + Na]^+^, calcd 621.499).

##### 1-(2-{[(3*β*)-3-Hydroxylup-20(29)-en-28-oyl]oxy}ethyl)-3-methylimidazolium bromide (**11a**) [EMIM-O-Lup][Br]

Yield: 0.50 g, 78%, white solid, mp 270–272 °C. [α]_D_^19^ + 2.9 (c 0.85, MeOH); IR (KBr) ν_max_ 2940, 2868, 1728, 1576, 1452, 1385, 1165, 1133, 1046, 879, 751, 621 cm^−1^; ^1^H NMR (DMSO-d_6_, 400 MHz) *δ* 9.41 (1H, s, Imid), 7.89 (1H, m, Imid), 7.83 (1H, m, Imid), 4.67 (1H, br s, H-29), 4.55 (1H, br s, H-29), 4.55 (2H, m, CH_2_N), 4.40 (2H, m, CH_2_O), 3.88 (3H, s, CH_3_N), 2.96 (1H, m, H-3), 2.78 (1H, m, H-19), 2.10–0.58 (24H, m), 1.62 (3H, s, H-30), 0.89 (3H, s, H-27), 0.86 (3H, s, H-23), 0.74 (3H, s, H-26), 0.71 (3H, s, H-25), 0.64 (3H, s, H-24); ^13^C NMR (DMSO-d_6_, 100 MHz) *δ* 175.1 (C-28), 150.4 (C-20), 137.5 (Imid), 124.2 (Imid), 123.0 (Imid), 110.3 (C-29), 77.2 (C-3), 62.3 (CH_2_O), 56.4 (C-17), 55.3 (C-5), 50.3 (C-9), 49.1 (C-18), 48.3 (CH_2_N), 47.0 (C-19), 42.3 (C-14), 40.6 (C-8), 38.9 (C-4), 38.7 (C-1), 38.1 (C-13), 37.1 (C-10), 36.5 (C-22), 36.3 (CH_3_N), 34.3 (C-7), 31.7 (C-16), 30.3 (C-21), 29.4 (C-15), 28.6 (C-23), 27.6 (C-2), 25.4 (C-12), 20.8 (C-11), 19.3 (C-30), 18.4 (C-6), 16.4 (C-25), 16.3 (C-24), 16.1 (C-26), 14.8 (C-27); anal. calcd for C_36_H_57_BrN_2_O_3_: C, 76.41; H, 10.15; found C, 76.29; H, 10.12. MALDI TOF: *m*/*z* 593.486 (calculated for cation C_38_H_61_N_2_O_3_ [M + Na]^+^, calcd 593.436).

##### 1-(2-{[(3*β*)-3-Hydroxylup-20(29)-en-28-oyl]oxy}butyl)-3-methylimidazolium bromide (**11b**) [BMIM-O-Lup][Br]

Yield: 0.58 g, 86%, white solid, mp 182–184 °C. [α]_D_^21^ + 0.4 (c 0.85, MeOH); IR (KBr) ν_max_ 2924, 2855, 1719, 1462, 1377, 1153, 1045, 973, 879, 743, 621 cm^−1^; ^1^H NMR (DMSO-d_6_, 400 MHz) *δ* 9.27 (1H, s, Imid), 7.82 (1H, m, Imid), 7.76 (1H, m, Imid), 4.68 (1H, br s, H-29), 4.56 (1H, br s, H-29), 4.24 (2H, t, *J* = 6.8 Hz, CH_2_N), 4.04 (2H, t, *J* = 6.0 Hz, CH_2_O), 3.87 (3H, s, CH_3_N), 2.97 (1H, m, H-3), 2.89 (1H, m, H-19), 2.18–0.60 (24H, m), 1.85 (2H, m, CH_2_), 1.64 (3H, s, H-30), 1.56 (2H, m, CH_2_), 0.92 (3H, s, H-27), 0.86 (3H, s, H-23), 0.81 (3H, s, H-26), 0.75 (3H, s, H-25), 0.61 (3H, s, H-24); ^13^C NMR (DMSO-d_6_, 100 MHz) *δ* 175.6 (C-28), 150.5 (C-20), 137.1 (Imid), 124.1 (Imid), 122.7 (Imid), 110.3 (C-29), 77.2 (C-3), 63.2 (CH_2_O), 56.3 (C-17), 55.3 (C-5), 50.4 (C-9), 49.1 (C-18), 48.7 (CH_2_N), 47.1 (C-19), 42.4 (C-14), 40.6 (C-8), 38.9 (C-4), 38.7 (C-1), 38.1 (C-13), 37.2 (C-10), 36.7 (C-22), 36.3 (CH_3_N), 34.3 (C-7), 31.9 (C-16), 30.5 (C-21), 29.6 (C-15), 28.6 (C-23), 27.6 (C-2), 26.8 (CH_2_), 25.5 (C-12, CH_2_), 20.8 (C-11), 19.4 (C-30), 18.4 (C-6), 16.4 (C-25), 16.3 (C-24), 16.1 (C-26), 14.8 (C-27); anal. calcd for C_38_H_61_BrN_2_O_3_: C, 67.74; H, 9.12; found C, 67.61; H, 9.09. MALDI TOF: *m*/*z* 593.558 (calculated for cation C_38_H_61_N_2_O_3_ [M + Na]^+^, calcd 593.468).

##### 1-(2-{[(3*β*)-3-Hydroxylup-20(29)-en-28-oyl]oxy}hexyl)-3-methylimidazolium bromide (**11c**) [HMIM-O-Lup][Br]

Yield: 0.65 g, 93%, white solid, mp 158–160 °C. [α]_D_^21^ + 0.6 (c 0.73, MeOH); IR (KBr) ν_max_ 2924, 2854, 1722, 1456, 1377, 1155, 1064, 973, 880, 724, 621 cm^−1^; ^1^H NMR (DMSO-d_6_, 400 MHz) *δ* 9.27 (1H, s, Imid), 7.82 (1H, m, Imid), 7.76 (1H, m, Imid), 4.68 (1H, br s, H-29), 4.56 (1H, br s, H-29), 4.15 (2H, t, *J* = 6.8 Hz, CH_2_N), 4.00 (2H, m, CH_2_O), 3.87 (3H, s, CH_3_N), 2.96 (1H, m, H-3), 2.89 (1H, m, H-19), 2.16–0.58 (24H, m), 1.78 (2H, m, CH_2_), 1.63 (3H, s, H-30), 1.55 (2H, m, CH_2_), 1.35 (2H, m, CH_2_), 1.25 (2H, m, CH_2_), 0.91 (3H, s, H-27), 0.85 (3H, s, H-23), 0.81 (3H, s, H-26), 0.73 (3H, s, H-25), 0.63 (3H, s, H-24); ^13^C NMR (DMSO-d_6_, 100 MHz) *δ* 175.7 (C-28), 150.5 (C-20), 136.9 (Imid), 124.0 (Imid), 122.7 (Imid), 110.2 (C-29), 77.3 (C-3), 63.8 (CH_2_O), 56.3 (C-17), 55.3 (C-5), 50.3 (C-9), 49.2 (CH_2_N), 49.1 (C-18), 47.2 (C-19), 42.4 (C-14), 40.7 (C-8), 38.9 (C-4), 38.7 (C-1), 38.2 (C-13), 37.1 (C-10), 36.8 (C-22), 36.2 (CH_3_N), 34.3 (C-7), 31.9 (C-16), 30.5 (C-21), 29.8 (CH_2_), 29.5 (C-15), 28.5 (C-23), 28.4 (CH_2_), 27.5 (C-2), 25.5 (C-12, CH_2_), 25.4 (CH_2_), 20.9 (C-11), 19.3 (C-30), 18.4 (C-6), 16.3 (C-25), 16.2 (C-24), 16.1 (C-26), 14.8 (C-27); anal. calcd for C_40_H_65_BrN_2_O_3_: C, 68.45; H, 9.33; found C, 68.29; H, 9.30. MALDI TOF: *m*/*z* 521.638 (calculated for cation C_40_H_65_N_2_O_3_ [M + Na]^+^, calcd 621.499).

#### 2.3.5. General Synthetic Procedure for Ionic Compounds (ICs) with Oleanolic Acid Covalently Linked to the Different Cations (**12a**–**12f**)

The method for the synthesis of ionic compounds of oleanolic acid **12a**–**12f** with different tertiary amines is similar to the previous method for the synthesis of ionic compounds (ICs) with triterpenoids covalently linked to the *N*-methylimidazole cation (see above Section 2.3.4).

##### 1-(6-{[(3*β*)-3-Hydroxyolean-12-en-28-oyl]oxy}hexyl)-pyridinium bromide (**12a**) [HPy-O-Olean][Br]

Yield: 0.61 g, 88%, white solid, mp 128–130 °C. [α]_D_^21^ + 30.5 (c 0.86, MeOH); IR (KBr) ν_max_ 2923, 2852, 2099, 1734, 1619, 1573, 1458, 1384, 1364, 1261, 1235, 1201, 1158, 1029, 754, 664, 621 cm^−1^; ^1^H NMR (DMSO-d_6_, 500 MHz) *δ* 9.09 (2H, d, *J* = 6.0 Hz, Py), 8.61 (1H, t, *J* = 7.5 Hz, Py), 8.15 (2H, t, *J* = 7.0 Hz, Py), 5.16 (1H, m, H-12), 4.59 (2H, t, *J* = 7.5 Hz, CH_2_N), 3.93 (2H, t, *J* = 6.5 Hz, CH_2_O), 2.99 (1H, m, H-3), 2.76 (1H, d, ^2^*J* = 10.0 Hz, H-18), 1.98–0.61 (22H, m), 1.90 (2H, m, CH_2_), 1.52 (2H, m, CH_2_), 1.33 (2H, m, CH_2_), 1.28 (2H, m, CH_2_), 1.07 (3H, s, H-27), 0.88 (3H, s, H-23), 0.86 (6H, s, H-29, H-30), 0.80 (3H, s, H-25), 0.66 (3H, s, H-24), 0.62 (3H, s, H-26); ^13^C NMR (DMSO-d_6_, 125 MHz) *δ* 177.1 (C-28), 145.9 (Py), 145.1 (Py), 143.9 (C-13), 128.6 (Py), 122.3 (C-12), 77.3 (C-3), 63.9 (CH_2_O), 61.2 (CH_2_N), 55.2 (C-5), 47.4 (C-9), 46.5 (C-17), 45.8 (C-19), 41.7 (C-14), 41.4 (C-18), 39.3 (C-8), 38.8 (C-4), 38.5 (C-1), 36.9 (C-10), 33.6 (C-21), 33.2 (C-29), 32.8 (C-7), 32.5 (C-22), 31.1 (CH_2_), 30.8 (C-20), 28.7 (C-23), 28.3 (CH_2_), 27.5 (C-15), 27.4 (C-2), 26.0 (C-27), 25.4 (CH_2_), 25.3 (CH_2_), 23.8 (C-30), 23.4 (C-11), 22.9 (C-16), 18.4 (C-6), 17.2 (C-26), 16.5 (C-24), 15.5 (C-25); anal. calcd for C_41_H_64_BrNO_3_: C, 70.46; H, 9.23; found C, 70.38; H, 9.20. MALDI TOF: *m*/*z* 618.441 (calculated for cation C_41_H_64_NO_3_ [M]^+^, calcd 618.952).

##### 1-(6-{[(3*β*)-3-Hydroxyolean-12-en-28-oyl]oxy}hexyl)-1-methylpiperidinium bromide (**12b**) [HMPip-O-Olean][Br]

Yield: 0.66 g, 92%, white solid, mp 138–140 °C. [α]_D_^21^ + 31.2 (c 0.69, MeOH); IR (KBr) ν_max_ 2945, 2866, 1719, 1654, 1464, 1386, 1262, 1177, 1084, 1034, 946, 882, 655 cm^−1^; ^1^H NMR (DMSO-d_6_, 400 MHz) *δ* 5.19 (1H, m, H-12), 4.30 (1H, d, *J* = 5.2 Hz, OH), 3.96 (2H, t, *J* = 6.0 Hz, CH_2_O), 3.33 (6H, m, CH_2_N), 3.01 (4H, m, CH_3_N, H-3), 2.79 (1H, d, ^2^*J* = 10.0 Hz, H-18), 2.00–0.64 (22H, m), 1.78 (4H, m, CH_2_CH_2_N), 1.65 (2H, m, CH_2_), 1.57 (2H, m, CH_2_), 1.55 (2H, m, CH_2_), 1.35 (2H, m, CH_2_), 1.30 (2H, m, CH_2_), 1.10 (3H, s, H-27), 0.89 (3H, s, H-23), 0.88 (6H, s, H-29, H-30), 0.85 (3H, s, H-25), 0.67 (3H, s, H-24), 0.66 (3H, s, H-26); ^13^C NMR (DMSO-d_6_, 100 MHz) *δ* 177.0 (C-28), 143.9 (C-13), 122.3 (C-12), 77.2 (C-3), 63.9 (CH_2_O), 62.6 (CH_2_N), 60.4 (CH_2_N, signals of 2C), 55.2 (C-5), 47.5 (C-9, CH_3_N), 46.5 (C-17), 45.9 (C-19), 41.7 (C-14), 41.4 (C-18), 39.3 (C-8), 38.8 (C-4), 38.5 (C-1), 37.0 (C-10), 33.7 (C-21), 33.2 (C-29), 32.9 (C-7), 32.6 (C-22), 30.8 (C-20), 28.7 (C-23), 28.3 (CH_2_), 27.7 (C-15), 27.4 (C-2), 26.0 (C-27), 25.9 (CH_2_), 25.5 (CH_2_), 23.8 (C-30), 23.4 (C-11), 23.0 (C-16), 21.4 (CH_2_), 21.2 (CH_2_), 19.8 (CH_2_CH_2_N, signals of 2C), 18.5 (C-6), 17.2 (C-26), 16.5 (C-24), 15.6 (C-25); anal. calcd for C_42_H_72_BrNO_3_: C, 70.17; H, 10.09; found C, 70.01; H, 10.05. MALDI TOF: *m*/*z* 638.594 (calculated for cation C_42_H_72_NO_3_ [M]^+^, calcd 638.551).

##### 1-(6-{[(3*β*)-3-Hydroxyolean-12-en-28-oyl]oxy}hexyl)-1-methylpyrrolidinium bromide (**12c**) [HMPyrr-O-Olean][Br]

Yield: 0.64 g, 91%, white solid, mp 136–138 °C. [α]_D_^22^ + 35.4 (c 0.81, MeOH); IR (KBr) ν_max_ 2944, 2864, 2718, 1719, 1620, 1461, 1385, 1363, 1261, 1178, 1161, 1048, 1032, 1009, 754, 665, 621 cm^−1^; ^1^H NMR (DMSO-d_6_, 500 MHz) *δ* 5.19 (1H, m, H-12), 3.96 (2H, m, CH_2_O), 3.48 (4H, m, CH_2_N), 3.30 (2H, m, CH_2_N), 2.99 (4H, m, H-3, CH_3_N), 2.79 (1H, d, ^2^*J* = 10.0 Hz, H-18), 2.09 (4H, m, CH_2_CH_2_N), 2.04–0.64 (22H, m), 1.75 (2H, m, CH_2_), 1.57 (2H, m, CH_2_), 1.35 (2H, m, CH_2_), 1.30 (2H, m, CH_2_), 1.10 (3H, s, H-27), 0.89 (3H, s, H-23), 0.88 (6H, s, H-29, H-30), 0.85 (3H, s, H-25), 0.67 (3H, s, H-24), 0.66 (3H, s, H-26); ^13^C NMR (DMSO-d_6_, 125 MHz) *δ* 177.1 (C-28), 143.9 (C-13), 122.3 (C-12), 77.2 (C-3), 63.9 (CH_2_O), 63.8 (CH_2_N, signals of 2C), 63.4 (CH_2_N), 55.2 (C-5), 47.9 (CH_3_N), 47.5 (C-9), 46.5 (C-17), 45.8 (C-19), 41.7 (C-14), 41.4 (C-18), 39.3 (C-8), 38.8 (C-4), 38.5 (C-1), 37.0 (C-10), 33.6 (C-21), 33.2 (C-29), 32.7 (C-7), 32.5 (C-22), 30.8 (C-20), 28.7 (C-23), 28.3 (CH_2_), 27.6 (C-15), 27.4 (C-2), 26.0 (C-27), 25.9 (CH_2_), 25.5 (CH_2_), 23.8 (CH_2_), 23.3 (C-11, C-30), 23.0 (C-16), 21.6 (CH_2_CH_2_N, signals of 2C), 18.5 (C-6), 17.2 (C-26), 16.5 (C-24), 15.6 (C-25); anal. calcd for C_41_H_70_BrNO_3_: C, 69.86; H, 10.01; found C, 69.78; H, 9.98. MALDI TOF: *m*/*z* 624.552 (calculated for cation C_41_H_64_NO_3_ [M]^+^, calcd 624.536).

##### 4-(6-{[(3*β*)-3-Hydroxyolean-12-en-28-oyl]oxy}hexyl)-4-methylmorpholinium bromide (**12d**) [HMMor-O-Olean][Br]

Yield: 0.64 g, 89%, white solid, mp 88–90 °C. [α]_D_^21^ + 34.2 (c 0.86, MeOH); IR (KBr) ν_max_ 2945, 2868, 1722, 1657, 1463, 1386, 1262, 1235, 1177, 1161, 1126, 1048, 891, 826, 668, 602 cm^−1^; ^1^H NMR (DMSO-d_6_, 400 MHz) *δ* 5.17 (1H, m, H-12), 4.30 (1H, d, *J* = 5.2 Hz, OH), 3.96–3.91 (6H, m, CH_2_O), 3.50–3.42 (6H, m, CH_2_N), 3.16 (3H, s, CH_3_N), 2.98 (1H, m, H-3), 2.77 (1H, d, ^2^*J* = 10.4 Hz, H-18), 2.00–0.64 (22H, m), 1.68 (2H, m, CH_2_), 1.58 (2H, m, CH_2_), 1.35 (2H, m, CH_2_), 1.30 (2H, m, CH_2_), 1.08 (3H, s, H-27), 0.88 (3H, s, H-23), 0.87 (6H, s, H-29, H-30), 0.83 (3H, s, H-25), 0.66 (3H, s, H-24), 0.65 (3H, s, H-26); ^13^C NMR (DMSO-d_6_, 100 MHz) *δ* 176.7 (C-28), 143.6 (C-13), 122.0 (C-12), 76.9 (C-3), 63.7 (CH_2_N, CH_2_O), 59.9 (CH_2_O, signals of 2C), 59.1 (CH_2_N, signals of 2C), 54.9 (C-5), 47.2 (C-9), 46.2 (C-17, CH_3_N), 45.5 (C-19), 41.4 (C-14), 41.1 (C-18), 39.0 (C-8), 38.5 (C-4), 38.2 (C-1), 36.7 (C-10), 33.4 (C-21), 32.9 (C-29), 32.6 (C-7), 32.3 (C-22), 30.5 (C-20), 28.4 (C-23), 28.0 (CH_2_), 27.3 (C-15), 27.1 (C-2), 25.7 (C-27), 25.5 (CH_2_), 25.2 (CH_2_), 23.5 (C-30), 23.1 (C-11), 22.7 (C-16), 20.9 (CH_2_), 18.2 (C-6), 16.9 (C-26), 16.2 (C-24), 15.3 (C-25); anal. calcd for C_41_H_70_BrNO_4_: C, 68.31; H, 9.79; found C, 68.19; H, 9.67. MALDI TOF: *m*/*z* 640.525 (calculated for cation C_41_H_70_NO_4_ [M]^+^, calcd 640.530).

##### N,N,N-Triethyl-N-(6-{[(3*β*)-3-hydroxyolean-12-en-28-oyl]oxy}hexyl)-aminium bromide (**12e**) [HTEA-O-Olean][Br]

Yield: 0.55 g, 76%, white solid, mp 136–138 °C. [α]_D_^19^ + 13.8 (c 0.81, MeOH); IR (KBr) ν_max_ 2928, 2870, 1704, 1641, 1487, 1466, 1389, 1362, 1262, 1178, 1069, 1031, 740, 665 cm^−1^; ^1^H NMR (DMSO-d_6_, 500 MHz) *δ* 5.18 (1H, m, H-12), 4.32 (1H, d, *J* = 5.0 Hz, OH), 3.95 (2H, m, CH_2_O), 3.24 (6H, q, *J* = 7.5 Hz, CH_2_N), 3.11 (2H, m, CH_2_N), 2.99 (1H, m, H-3), 2.70 (1H, d, ^2^*J* = 10.0 Hz, H-18), 2.00–0.66 (22H, m), 1.56 (4H, m, CH_2_), 1.37 (2H, m, CH_2_), 1.30 (2H, m, CH_2_), 1.16 (9H, t, *J* = 7.5 Hz, CH_3_CH_2_N), 1.09 (3H, s, H-27), 0.89 (3H, s, H-23), 0.88 (6H, s, H-29, H-30), 0.84 (3H, s, H-25), 0.67 (3H, s, H-24), 0.66 (3H, s, H-26); ^13^C NMR (DMSO-d_6_, 125 MHz) *δ* 177.1 (C-28), 143.9 (C-13), 122.3 (C-12), 77.2 (C-3), 64.0 (CH_2_O), 56.4 (CH_2_N), 55.2 (C-5), 52.5 (CH_2_N, signals of 3C), 47.5 (C-9), 46.6 (C-17), 45.8 (C-19), 41.7 (C-14), 41.4 (C-18), 39.3 (C-8), 38.8 (C-4), 38.5 (C-1), 37.0 (C-10), 33.6 (C-21), 33.2 (C-29), 32.9 (C-7), 32.6 (C-22), 30.8 (C-20), 28.7 (C-23), 28.4 (CH_2_), 27.6 (C-15), 27.4 (C-2), 26.0 (C-27), 25.9 (CH_2_), 25.5 (CH_2_), 23.8 (C-30), 23.4 (C-11), 23.0 (C-16), 21.4 (CH_2_), 18.4 (C-6), 17.2 (C-26), 16.5 (C-24), 15.6 (C-25), 7.7 (CH_3_, signals of 3C); anal. calcd for C_42_H_74_BrNO_3_: C, 69.97; H, 10.35; found C, 69.81; H, 10.33. MALDI TOF: *m*/*z* 640.551 (calculated for cation C_42_H_74_NO_3_ [M]^+^, calcd 640.567).

##### N-(2-Hydroxyethyl)-N-(6-{[(3*β*)-3-hydroxyolean-12-en-28-oyl]oxy}hexyl)-N,N-dimethylaminium bromide (**12f**) [HChol-O-Olean][Br]

Yield: 0.62 g, 88%, white solid, mp 110–112 °C. [α]_D_^21^ + 31.9 (c 0.88, MeOH); IR (KBr) ν_max_ 2929, 2866, 1722, 1463, 1385, 1302, 1262, 1178, 1161, 1085, 1048, 880, 755, 655 cm^−1^; ^1^H NMR (DMSO-d_6_, 400 MHz) *δ* 5.28 (1H, t, *J* = 5.0 Hz, OH), 5.17 (1H, m, H-12), 4.30 (1H, d, *J* = 4.8 Hz, OH), 3.94 (2H, t, *J* = 6.2 Hz, CH_2_O), 3.82 (2H, m, CH_2_OH), 3.42 (2H, t, *J* = 4.8 Hz, CH_2_N), 3.37 (2H, m, CH_2_N), 3.09 (6H, s, CH_3_N), 2.99 (1H, m, H-3), 2.77 (1H, d, ^2^*J* = 10.0 Hz, H-18), 2.00–0.64 (22H, m), 1.68 (2H, m, CH_2_), 1.56 (2H, m, CH_2_), 1.35 (2H, m, CH_2_), 1.27 (2H, m, CH_2_), 1.09 (3H, s, H-27), 0.88 (3H, s, H-23), 0.87 (6H, s, H-29, H-30), 0.84 (3H, s, H-25), 0.66 (3H, s, H-24), 0.65 (3H, s, H-26); ^13^C NMR (DMSO-d_6_, 100 MHz) *δ* 176.9 (C-28), 143.9 (C-13), 122.3 (C-12), 77.2 (C-3), 65.0 (CH_2_N), 64.4 (CH_2_N), 63.9 (CH_2_O), 55.3 (CH_2_OH, C-5), 51.3 (CH_3_, signals of 2C), 47.5 (C-9), 46.5 (C-17), 45.9 (C-19), 41.7 (C-14), 41.4 (C-18), 39.3 (C-8), 38.8 (C-4), 38.6 (C-1), 37.0 (C-10), 33.7 (C-21), 33.2 (C-29), 32.9 (C-7), 32.6 (C-22), 30.8 (C-20), 28.7 (C-23), 28.3 (CH_2_), 27.6 (C-15), 27.4 (C-2), 26.1 (C-27), 25.9 (CH_2_), 25.5 (CH_2_), 23.8 (C-30), 23.4 (C-11), 23.0 (C-16), 22.2 (CH_2_), 18.5 (C-6), 17.2 (C-26), 16.5 (C-24), 15.6 (C-25); anal. calcd for C_40_H_70_BrNO_4_: C, 67.77; H, 9.95; found C, 67.58; H, 9.92. MALDI TOF: *m*/*z* 628.375 (calculated for cation C_40_H_70_NO_4_ [M]^+^, calcd 628.530).

#### 2.3.6. General Synthetic Procedure for Ionic Compounds (ICs) of Oleanolic Acid with Different Anions (**13a**–**13g**)

1-(6-{[(3*β*)-3-Hydroxyolean-12-en-28-oyl]oxy}hexyl)-3-methylimidazolium bromide **9c** [HMIM-O-Olean][Br] (0.5 mmol, 0.35 g), sodium tetrafluoroborate (0.75 mmol, 82 mg) and 5 mL of dry MeOH were placed into a flask, flushed with argon and stirred at room temperature for 24 h. After the end of the reaction, the reaction mixture was filtered off from the solid residue and the solvent was distilled off on a rotary evaporator. The product of reaction was purified by column chromatography using CHCl_3_/MeOH (10/1) as the elution solvent. Ionic compounds (ICs) of oleanolic acid with other anions were obtained using a similar procedure.

##### 1-(6-{[(3*β*)-3-Hydroxyolean-12-en-28-oyl]oxy}hexyl)-3-methylimidazolium Tetrafluoroborate (**13a**) [HMIM-O-Olean][BF_4_]

Yield: 0.68 g, 96%, white solid, mp 108–110 °C. [α]_D_^25^ + 42.1 (c 0.80, MeOH); IR (KBr) ν_max_ 2929, 2863, 1715, 1621, 1573, 1456, 1385, 1302, 1261, 1177, 1161, 1047, 769, 665, 624*δ* 522 cm^−1^; ^1^H NMR (DMSO-d_6_, 500 MHz) *δ* 9.06 (1H, s, Imid), 7.74 (1H, d, *J* = 2.0 Hz, Imid), 7.68 (1H, d, *J* = 1.5 Hz, Imid), 5.17 (1H, m, H-12), 4.35 (1H, br d, *J* = 4.5 Hz, OH), 4.14 (2H, t, *J* = 7.5 Hz, CH_2_N), 3.93 (2H, t, *J* = 6.5 Hz, CH_2_O), 3.84 (3H, s, NCH_3_), 2.99 (1H, m, H-3), 2.77 (1H, d, ^2^*J* = 9.5 Hz, H-18), 1.99–0.63 (22H, m), 1.77 (2H, m, CH_2_), 1.55 (2H, m, CH_2_), 1.35 (2H, m, CH_2_), 1.25 (2H, m, CH_2_), 1.09 (3H, s, H-27), 0.89 (3H, s, H-23), 0.88 (6H, s, H-29, H-30), 0.83 (3H, s, H-25), 0.67 (3H, s, H-24), 0.65 (3H, s, H-26); ^13^C NMR (DMSO-d_6_, 125 MHz) *δ* 177.1 (C-28), 143.9 (C-13), 136.9 (Imid), 124.0 (Imid), 122.7 (Imid), 122.3 (C-12), 77.3 (C-3), 64.0 (CH_2_O), 55.2 (C-5), 49.2 (CH_2_N), 47.5 (C-9), 46.5 (C-17), 45.9 (C-19), 41.7 (C-14), 41.4 (C-18), 39.3 (C-8), 38.8 (C-4), 38.5 (C-1), 36.9 (C-10), 36.2 (CH_3_N), 33.6 (C-21), 33.2 (C-29), 32.8 (C-7), 32.6 (C-22), 30.8 (C-20), 29.8 (CH_2_), 28.7 (C-23), 28.3 (CH_2_), 27.5 (C-15), 27.4 (C-2), 26.0 (C-27), 25.6 (CH_2_), 25.4 (CH_2_), 23.8 (C-30), 23.4 (C-11), 23.0 (C-16), 18.4 (C-6), 17.2 (C-26), 16.5 (C-24), 15.5 (C-25); ^19^F NMR (DMSO-d_6_, 376 MHz) *δ* -148.3; anal. calcd for C_40_H_65_BF_4_N_2_O_3_: C, 67.78; H, 9.24; found C, 67.69; H, 9.22. MALDI TOF: *m*/*z* 621.496 (calculated for cation C_40_H_65_N_2_O_3_ [M]^+^, calcd 621.499).

##### 1-(6-{[(3*β*)-3-Hydroxyolean-12-en-28-oyl]oxy}hexyl)-3-methylimidazolium Hexafluoroantimonate (**13b**) [HMIM-O-Olean][SbF_6_]

Yield: 0.83 g, 97%, white solid, mp 70–72 °C. [α]_D_^19^ + 22.1 (c 0.76, MeOH); IR (KBr) ν_max_ 2924, 2854, 1723, 1648, 1574, 1469, 1386, 1303, 1262, 1163, 1032, 844, 756, 659, 623, 567 cm^−1^; ^1^H NMR (DMSO-d_6_, 400 MHz) *δ* 9.10 (1H, s, Imid), 7.75 (1H, m, Imid), 7.70 (1H, m, Imid), 5.18 (1H, m, H-12), 4.29 (1H, br d, *J* = 3.6 Hz, OH), 4.16 (2H, t, *J* = 6.8 Hz, CH_2_N), 3.95 (2H, t, *J* = 6.4 Hz, CH_2_O), 3.85 (3H, s, NCH_3_), 3.00 (1H, m, H-3), 2.79 (1H, d, ^2^*J* = 9.6 Hz, H-18), 2.00–0.63 (22H, m), 1.78 (2H, m, CH_2_), 1.55 (2H, m, CH_2_), 1.35 (2H, m, CH_2_), 1.25 (2H, m, CH_2_), 1.10 (3H, s, H-27), 0.90 (3H, s, H-23), 0.88 (6H, s, H-29, H-30), 0.82 (3H, s, H-25), 0.69 (3H, s, H-24), 0.66 (3H, s, H-26); ^13^C NMR (DMSO-d_6_, 100 MHz) *δ* 177.0 (C-28), 143.9 (C-13), 136.9 (Imid), 124.1 (Imid), 122.7 (Imid), 122.3 (C-12), 77.3 (C-3), 63.9 (CH_2_O), 55.2 (C-5), 49.2 (CH_2_N), 47.5 (C-9), 46.5 (C-17), 45.9 (C-19), 41.7 (C-14), 41.4 (C-18), 39.3 (C-8), 38.8 (C-4), 38.5 (C-1), 37.0 (C-10), 36.2 (CH_3_N), 33.6 (C-21), 33.2 (C-29), 32.9 (C-7), 32.6 (C-22), 30.8 (C-20), 29.8 (CH_2_), 28.7 (C-23), 28.3 (CH_2_), 27.6 (C-15), 27.4 (C-2), 26.0 (C-27), 25.6 (CH_2_), 25.4 (CH_2_), 23.8 (C-30), 23.4 (C-11), 23.0 (C-16), 18.5 (C-6), 17.2 (C-26), 16.5 (C-24), 15.5 (C-25); ^19^F NMR (DMSO-d_6_, 376 MHz) *δ* -17.9; anal. calcd for C_40_H_65_F_6_N_2_O_3_Sb: C, 56.01; H, 7.64; found C, 55.88; H, 7.61. MALDI TOF: *m*/*z* 621.476 (calculated for cation C_40_H_65_N_2_O_3_ [M]^+^, calcd 621.499).

##### 1-(6-{[(3*β*)-3-Hydroxyolean-12-en-28-oyl]oxy}hexyl)-3-methylimidazolium Hexafluorophosphate (**13c**) [HMIM-O-Olean][PF_6_]

Yield: 0.74 g, 97%, white solid, mp 78–80 °C. [α]_D_^21^ + 31.5 (c 0.79, MeOH); IR (KBr) ν_max_ 2926, 2856, 1704, 1634, 1575, 1463, 1386, 1266, 1167, 1084, 1021, 852, 740, 655, 624, 559 cm^−1^; ^1^H NMR (DMSO-d_6_, 400 MHz) *δ* 9.09 (1H, s, Imid), 7.76 (1H, m, Imid), 7.70 (1H, m, Imid), 5.18 (1H, m, H-12), 4.29 (1H, m, OH), 4.16 (2H, m, CH_2_N), 3.95 (2H, m, CH_2_O), 3.85 (3H, s, NCH_3_), 3.01 (1H, m, H-3), 2.79 (1H, d, ^2^*J* = 9.6 Hz, H-18), 2.00–0.63 (22H, m), 1.78 (2H, m, CH_2_), 1.55 (2H, m, CH_2_), 1.35 (2H, m, CH_2_), 1.25 (2H, m, CH_2_), 1.10 (3H, s, H-27), 0.90 (3H, s, H-23), 0.88 (6H, s, H-29, H-30), 0.84 (3H, s, H-25), 0.68 (3H, s, H-24), 0.66 (3H, s, H-26); ^13^C NMR (DMSO-d_6_, 100 MHz) *δ* 177.0 (C-28), 143.9 (C-13), 136.9 (Imid), 124.1 (Imid), 122.7 (Imid), 122.3 (C-12), 77.3 (C-3), 63.9 (CH_2_O), 55.2 (C-5), 49.2 (CH_2_N), 47.5 (C-9), 46.5 (C-17), 45.9 (C-19), 41.7 (C-14), 41.4 (C-18), 39.4 (C-8), 38.8 (C-4), 38.5 (C-1), 37.0 (C-10), 36.2 (CH_3_N), 33.7 (C-21), 33.2 (C-29), 32.9 (C-7), 32.6 (C-22), 30.8 (C-20), 29.8 (CH_2_), 28.7 (C-23), 28.3 (CH_2_), 27.6 (C-15), 27.4 (C-2), 26.0 (C-27), 25.6 (CH_2_), 25.4 (CH_2_), 23.8 (C-30), 23.4 (C-11), 23.0 (C-16), 18.5 (C-6), 17.2 (C-26), 16.5 (C-24), 15.5 (C-25); ^19^F NMR (DMSO-d_6_, 376 MHz) *δ* -69.2, -71.1; anal. calcd for C_40_H_65_F_6_N_2_O_3_Sb: C, 56.01; H, 7.64; found C, 55.88; H, 7.61. MALDI TOF: *m*/*z* 621.476 (calculated for cation C_40_H_65_N_2_O_3_ [M]^+^, calcd 621.499).

##### 1-(6-{[(3*β*)-3-Hydroxyolean-12-en-28-oyl]oxy}hexyl)-3-methylimidazolium Acetate (**13d**) [HMIM-O-Olean][CH_3_COO]

Yield: 0.65 g, 95%, white solid, mp 106–108 °C. [α]_D_^21^ + 28.6 (c 0.85, MeOH); IR (KBr) ν_max_ 2946, 2860, 1702, 1648, 1573, 1462, 1388, 1303, 1263, 1170, 1024, 755, 655, 623 cm^−1^; ^1^H NMR (DMSO-d_6_, 400 MHz) *δ* 9.15 (1H, s, Imid), 7.76 (1H, m, Imid), 7.69 (1H, m, Imid), 5.16 (1H, m, H-12), 4.16 (2H, t, *J* = 6.8 Hz, CH_2_N), 3.93 (2H, t, *J* = 6.0 Hz, CH_2_O), 3.85 (3H, s, NCH_3_), 2.99 (1H, m, H-3), 2.77 (1H, d, ^2^*J* = 10.8 Hz, H-18), 1.99–0.63 (22H, m), 1.89 (3H, s, COCH_3_), 1.77 (2H, m, CH_2_), 1.55 (2H, m, CH_2_), 1.35 (2H, m, CH_2_), 1.25 (2H, m, CH_2_), 1.07 (3H, s, H-27), 0.88 (3H, s, H-23), 0.86 (6H, s, H-29, H-30), 0.81 (3H, s, H-25), 0.66 (3H, s, H-24), 0.63 (3H, s, H-26); ^13^C NMR (DMSO-d_6_, 100 MHz) *δ* 177.1 (C-28), 173.0 (COCH_3_), 143.9 (C-13), 136.9 (Imid), 124.0 (Imid), 122.7 (Imid), 122.3 (C-12), 77.3 (C-3), 64.0 (CH_2_O), 55.2 (C-5), 49.2 (CH_2_N), 47.5 (C-9), 46.5 (C-17), 45.9 (C-19), 41.7 (C-14), 41.4 (C-18), 39.3 (C-8), 38.8 (C-4), 38.5 (C-1), 36.9 (C-10), 36.2 (CH_3_N), 33.6 (C-21), 33.2 (C-29), 32.8 (C-7), 32.6 (C-22), 30.8 (C-20), 29.8 (CH_2_), 28.7 (C-23), 28.3 (CH_2_), 27.5 (C-15), 27.4 (C-2), 26.0 (C-27), 25.6 (CH_2_), 25.4 (CH_2_), 23.8 (C-30), 23.4 (C-11), 23.0 (C-16), 21.8 (COCH_3_), 18.4 (C-6), 17.2 (C-26), 16.5 (C-24), 15.5 (C-25); anal. calcd for C_42_H_68_N_2_O_5_: C, 74.07; H, 10.06; found C, 73.68; H, 10.03. MALDI TOF: *m*/*z* 621.483 (calculated for cation C_40_H_65_N_2_O_3_ [M]^+^, calcd 621.499).

##### 1-(6-{[(3*β*)-3-Hydroxyolean-12-en-28-oyl]oxy}hexyl)-3-methylimidazolium Benzenesulfonate (**13e**) [HMIM-O-Olean][C_6_H_5_SO_3_]

Yield: 0.76 g, 97%, white solid, mp 70–72 °C. [α]_D_^21^ + 52.1 (c 0.86, MeOH); IR (KBr) ν_max_ 2943, 2865, 1721, 1657, 1573, 1463, 1386, 1177, 1128, 1039, 1019, 880, 761, 730, 691, 614 cm^−1^; ^1^H NMR (DMSO-d_6_, 400 MHz) *δ* 9.17 (1H, s, Imid), 7.78 (1H, m, Imid), 7.72 (1H, m, Imid), 7.62, 7.31 (5H, m, Ph), 5.18 (1H, m, H-12), 4.31 (1H, d, *J* = 4.8 Hz, OH), 4.15 (2H, t, *J* = 7.2 Hz, CH_2_N), 3.94 (2H, t, *J* = 6.4 Hz, CH_2_O), 3.85 (3H, s, NCH_3_), 2.99 (1H, m, H-3), 2.79 (1H, d, ^2^*J* = 10.0 Hz, H-18), 2.00–0.63 (22H, m), 1.78 (2H, m, CH_2_), 1.54 (2H, m, CH_2_), 1.35 (2H, m, CH_2_), 1.25 (2H, m, CH_2_), 1.10 (3H, s, H-27), 0.90 (3H, s, H-23), 0.88 (6H, s, H-29, H-30), 0.84 (3H, s, H-25), 0.68 (3H, s, H-24), 0.66 (3H, s, H-26); ^13^C NMR (DMSO-d_6_, 100 MHz) *δ* 177.0 (C-28), 148.8 (Ph), 143.9 (C-13), 137.0 (Imid), 128.9 (Ph, signals 2C), 128.1 (Ph, signals 2C), 125.9 (Ph), 124.1 (Imid), 122.7 (Imid), 122.3 (C-12), 77.3 (C-3), 64.0 (CH_2_O), 55.3 (C-5), 49.1 (CH_2_N), 47.5 (C-9), 46.5 (C-17), 45.9 (C-19), 41.7 (C-14), 41.4 (C-18), 39.3 (C-8), 38.8 (C-4), 38.5 (C-1), 37.0 (C-10), 36.2 (CH_3_N), 33.7 (C-21), 33.2 (C-29), 32.9 (C-7), 32.6 (C-22), 30.8 (C-20), 29.8 (CH_2_), 28.7 (C-23), 28.3 (CH_2_), 27.6 (C-15), 27.4 (C-2), 26.1 (C-27), 25.6 (CH_2_), 25.4 (CH_2_), 23.8 (C-30), 23.4 (C-11), 23.0 (C-16), 18.5 (C-6), 17.2 (C-26), 16.5 (C-24), 15.5 (C-25); anal. calcd for C_46_H_70_N_2_O_6_S: C, 70.91; H, 9.06; found C, 70.81; H, 9.01. MALDI TOF: *m*/*z* 621.463 (calculated for cation C_40_H_65_N_2_O_3_ [M]^+^, calcd 621.499).

##### 1-(6-{[(3*β*)-3-Hydroxyolean-12-en-28-oyl]oxy}hexyl)-3-methylimidazolium Salicylate (**13f**) [HMIM-O-Olean][Sal]

Yield: 0.73 g, 97%, white solid, mp 78–80 °C. [α]_D_^18^ + 40.0 (c 0.86, MeOH); IR (KBr) ν_max_ 2929, 2861, 1718, 1660, 1612, 1573, 1463, 1386, 1254, 1161, 1031, 864, 756, 663, 623, 531 cm^−1^; ^1^H NMR (DMSO-d_6_, 400 MHz) *δ* 9.26 (1H, s, Imid), 7.81 (1H, m, Imid), 7.78 (1H, dd, *J* = 7.6 Hz, *J* = 1.6 Hz, Sal), 7.75 (1H, m, Imid), 7.44 (1H, td, *J* = 7.6 Hz, *J* = 1.6 Hz, Sal), 6.89 (1H, d, *J* = 7.6 Hz, Sal), 6.86 (1H, t, *J* = 7.2 Hz, Sal), 5.17 (1H, m, H-12), 4.17 (2H, t, *J* = 6.8 Hz, CH_2_N), 3.93 (2H, t, *J* = 6.0 Hz, CH_2_O), 3.87 (3H, s, NCH_3_), 2.99 (1H, m, H-3), 2.77 (1H, d, ^2^*J* = 9.6 Hz, H-18), 2.00–0.63 (22H, m), 1.78 (2H, m, CH_2_), 1.54 (2H, m, CH_2_), 1.35 (2H, m, CH_2_), 1.25 (2H, m, CH_2_), 1.08 (3H, s, H-27), 0.89 (3H, s, H-23), 0.87 (6H, s, H-29, H-30), 0.82 (3H, s, H-25), 0.67 (3H, s, H-24), 0.64 (3H, s, H-26); ^13^C NMR (DMSO-d_6_, 100 MHz) *δ* 177.0 (C-28), 172.5 (Sal), 161.8 (Sal), 143.9 (C-13), 137.0 (Imid), 135.3 (Sal), 130.7 (Sal), 124.1 (Imid), 122.7 (Imid), 122.3 (C-12), 119.1 (Sal), 117.3 (Sal), 114.7 (Sal), 77.2 (C-3), 63.9 (CH_2_O), 55.2 (C-5), 49.1 (CH_2_N), 47.5 (C-9), 46.5 (C-17), 45.8 (C-19), 41.7 (C-14), 41.4 (C-18), 39.3 (C-8), 38.8 (C-4), 38.5 (C-1), 37.0 (C-10), 36.2 (CH_3_N), 33.6 (C-21), 33.2 (C-29), 32.8 (C-7), 32.6 (C-22), 30.8 (C-20), 29.8 (CH_2_), 28.7 (C-23), 28.3 (CH_2_), 27.5 (C-15), 27.4 (C-2), 26.0 (C-27), 25.6(CH_2_), 25.4 (CH_2_), 23.8 (C-30), 23.4 (C-11), 23.0 (C-16), 18.4 (C-6), 17.2 (C-26), 16.5 (C-24), 15.5 (C-25); anal. calcd for C_47_H_70_N_2_O_6_: C, 74.37; H, 9.30; found C, 74.21; H, 9.27. MALDI TOF: *m*/*z* 621.459 (calculated for cation C_40_H_65_N_2_O_3_ [M]^+^, calcd 621.499).

##### 1-(6-{[(3*β*)-3-Hydroxyolean-12-en-28-oyl]oxy}hexyl)-3-methylimidazolium L-lactate (**13g**) [HMIM-O-Olean][(L)-Lac]

Yield: 0.68 g, 96%, white solid, mp 96–98 °C. [α]_D_^18^ + 34.4 (c 0.88, MeOH); IR (KBr) ν_max_ 2925, 2855, 1716, 1653, 1572, 1456, 1385, 1261, 1159, 1093, 1032, 847, 759, 661, 623, 559 cm^−1^; ^1^H NMR (DMSO-d_6_, 400 MHz) *δ* 9.25 (1H, s, Imid), 7.81 (1H, m, Imid), 7.75 (1H, m, Imid), 5.17 (1H, m, H-12), 4.30 (1H, d, *J* = 4.8 Hz, OH), 4.18 (2H, t, *J* = 7.2 Hz, CH_2_N), 3.94 (2H, t, *J* = 6.4 Hz, CH_2_O), 3.87 (3H, s, NCH_3_), 3.44 (1H, m, CH), 3.00 (1H, m, H-3), 2.77 (1H, d, ^2^*J* = 10.0 Hz, H-18), 2.00–0.63 (22H, m), 1.78 (2H, m, CH_2_), 1.54 (2H, m, CH_2_), 1.35 (2H, m, CH_2_), 1.25 (2H, m, CH_2_), 1.09 (3H, s, H-27), 1.05 (3H, t, *J* = 7.2 Hz, CH_3_), 0.89 (3H, s, H-23), 0.87 (6H, s, H-29, H-30), 0.83 (3H, s, H-25), 0.67 (3H, s, H-24), 0.65 (3H, s, H-26); ^13^C NMR (DMSO-d_6_, 100 MHz) *δ* 177.0 (C-28, Lac), 143.9 (C-13), 137.0 (Imid), 124.1 (Imid), 122.7 (Imid), 122.3 (C-12), 77.2 (C-3), 63.9 (CH_2_O), 56.5 (Lac), 55.2 (C-5), 49.1 (CH_2_N), 47.5 (C-9), 46.5 (C-17), 45.8 (C-19), 41.7 (C-14), 41.4 (C-18), 39.3 (C-8), 38.8 (C-4), 38.5 (C-1), 37.0 (C-10), 36.2 (CH_3_N), 33.6 (C-21), 33.2 (C-29), 32.8 (C-7), 32.6 (C-22), 30.8 (C-20), 29.8 (CH_2_), 28.7 (C-23), 28.3 (CH_2_), 27.5 (C-15), 27.4 (C-2), 26.0 (C-27), 25.6(CH_2_), 25.4 (CH_2_), 23.8 (C-30), 23.4 (C-11), 23.0 (C-16), 19.0 (Lac), 18.4 (C-6), 17.2 (C-26), 16.5 (C-24), 15.5 (C-25); anal. calcd for C_43_H_70_N_2_O_6_: C, 72.64; H, 9.92; found C, 72.49; H, 9.89. MALDI TOF: *m*/*z* 621.457 (calculated for cation C_40_H_65_N_2_O_3_ [M]^+^, calcd 621.499).

#### 2.3.7. General Synthetic Procedure for Ionic Compounds (ICs) Bearing Triterpenoids in Their Anion (**15a**–**15d**), (**16a**–**16d**), (**17a**–**17d**)

Sodium (3*β*)-3-hydroxyolean-12-en-28-oate **14** (0.24 g, 0.5 mmol), 1-alkyl-3-methylimidazolium bromide (0.5 mmol) and 10 mL of dry MeOH were flushed with argon and stirred for 24 h at room temperature. After the end of the reaction, the white precipitate was filtered off and the solvent was distilled off on a rotary evaporator. The product was dissolved in a small amount dry EtOH and CH_2_Cl_2_ and was filtered through a plug of celite from solid residues and starting materials; the solvent was distilled off on a rotary evaporator. Then, the procedure was repeated; the product was dried under vacuum at 60 °C for 4 h and then for 12 h in a desiccator over P_2_O_5_.

1-Alkyl-3-methylimidazolium (3*β*)-3-hydroxyurs-12-en-28-oate (**16a**–**16d**) and 1-alkyl-3-methylimidazolium (3*β*)-3-hydroxylup-20(29)-en-28-oate (**17a**–**17d**) were prepared with a similar procedure.

##### 1-Ethyl-3-methylimidazolium (3*β*)-3-hydroxyolean-12-en-28-oylate (**15a**) [EMIM][Olean]

Yield: 0.53 g, 94%, white solid, mp 290–292 °C. [α]_D_^16^ + 45.5 (c 0.84, MeOH); IR (KBr) ν_max_ 2925, 2852, 1683, 1631, 1557, 1461, 1385, 1305, 1242, 1167, 1029, 997, 738, 619 cm^−1^; ^1^H NMR (DMSO-d_6_, 400 MHz) *δ* 9.31 (1H, s, Imid), 7.80 (1H, m, Imid), 7.72 (1H, m, Imid), 5.04 (1H, m, H-12), 4.21 (2H, q, *J* = 7.2 Hz, CH_2_N), 3.86 (3H, s, NCH_3_), 2.99 (1H, br t, *J* = 6.4 Hz, H-3), 2.85 (1H, d, ^2^*J* = 10.4 Hz, H-18), 1.86–0.62 (22H, m), 1.42 (3H, t, *J* = 7.6 Hz, CH_3_), 1.06 (3H, s, H-27), 0.89 (3H, s, H-23), 0.87 (3H, s, H-30), 0.84 (6H, s, H-25, H-29), 0.72 (3H, s, H-26), 0.67 (3H, s, H-24); ^13^C NMR (DMSO-d_6_, 100 MHz) *δ* 180.9 (C-28), 146.2 (C-13), 136.9 (Imid), 124.0 (Imid), 122.4 (Imid), 120.4 (C-12), 77.3 (C-3), 55.4 (C-5), 47.8 (C-9), 47.2 (C-19), 46.0 (C-17), 44.6 (CH_2_N), 42.0 (C-18), 41.9 (C-14), 39.3 (C-8), 38.8 (C-4), 38.6 (C-1), 37.1 (C-10), 36.2 (CH_3_N), 34.6 (C-21), 33.7 (C-29), 33.3 (C-7), 33.1 (C-22), 31.1 (C-20), 28.7 (C-23), 28.1 (C-15), 27.4 (C-2), 26.1 (C-27), 24.1 (C-30), 23.8 (C-16), 23.4 (C-11), 18.6 (C-6), 17.7 (C-26), 16.5 (C-24), 15.6 (C-25, CH_3_); anal. calcd for C_36_H_58_N_2_O_3_: C, 76.28; H, 10.31; found C, 76.06; H, 10.28. MALDI TOF: *m*/*z* 478.302 (calculated for anion C_30_H_47_O_3_ [M + Na]^+^, calcd 478.682).

##### 1-Butyl-3-methylimidazolium (3*β*)-3-hydroxyolean-12-en-28-oylate (**15b**) [BMIM][Olean]

Yield: 0.58 g, 97%, white solid, mp 251–253 °C. [α]_D_^16^ + 46.5 (c 0.78, MeOH); IR (KBr) ν_max_ 2931, 2871, 1651 1636, 1567, 1462, 1386, 1308, 1260, 1168, 1089, 1030, 950, 753, 623 cm^−1^; ^1^H NMR (DMSO-d_6_, 400 MHz) *δ* 9.30 (1H, s, Imid), 7.79 (1H, m, Imid), 7.73 (1H, m, Imid), 5.02 (1H, m, H-12), 4.18 (2H, m, CH_2_N), 3.87 (3H, s, NCH_3_), 2.99 (1H, m, H-3), 2.89 (1H, d, ^2^*J* = 10.4 Hz, H-18), 1.90–0.60 (22H, m), 1.76 (2H, m, CH_2_), 1.25 (2H, m, CH_2_), 1.05 (3H, s, H-27), 0.89 (3H, s, H-23), 0.87 (3H, s, H-30), 0.85 (6H, s, H-25, H-29), 0.89 (3H, m, CH_3_), 0.73 (3H, s, H-26), 0.67 (3H, s, H-24); ^13^C NMR (DMSO-d_6_, 100 MHz) *δ* 181.7 (C-28), 146.7 (C-13), 137.1 (Imid), 124.0 (Imid), 122.7 (Imid), 120.1 (C-12), 77.3 (C-3), 55.4 (C-5), 48.9 (CH_2_N), 47.8 (C-9), 47.4 (C-19), 46.1 (C-17), 42.3 (C-18), 41.9 (C-14), 39.2 (C-8), 38.8 (C-4), 38.6 (C-1), 37.1 (C-10), 36.2 (CH_3_N), 34.8 (C-21), 33.8 (C-29), 33.5 (C-7), 33.2 (C-22), 31.9 (CH_2_), 31.2 (C-20), 28.7 (C-23), 28.1 (C-15), 27.4 (C-2), 26.1 (C-27), 24.2 (C-30), 23.9 (C-16), 23.4 (C-11), 19.2 (CH_2_), 18.6 (C-6), 17.8 (C-26), 16.5 (C-24), 15.6 (C-25), 13.7 (CH_3_); anal. calcd for C_38_H_62_N_2_O_3_: C, 76.72; H, 10.50; found C, 76.56; H, 10.48.

##### 1-Hexyl-3-methylimidazolium (3*β*)-3-hydroxyolean-12-en-28-oylate (**15c**) [HMIM][Olean]

Yield: 0.59 g, 96%, white solid, mp 196–198 °C. [α]_D_^16^ + 32.0 (c 0.74, MeOH); IR (KBr) ν_max_ 2929, 2859, 1650, 1634, 1568, 1457, 1374, 1308, 1260, 1167, 1087, 1047, 950, 752, 623 cm^−1^; ^1^H NMR (DMSO-d_6_, 400 MHz) *δ* 9.41 (1H, s, Imid), 7.84 (1H, m, Imid), 7.77 (1H, m, Imid), 5.03 (1H, m, H-12), 4.19 (2H, t, *J* = 7.2 Hz, CH_2_N), 3.88 (3H, s, NCH_3_), 2.99 (1H, m, H-3), 2.87 (1H, d, ^2^*J* = 10.4 Hz, H-18), 1.90–0.62 (22H, m), 1.78 (2H, m, CH_2_), 1.26 (6H, m, CH_2_), 1.05 (3H, s, H-27), 0.89 (3H, s, H-23), 0.87 (3H, s, H-30), 0.85 (6H, s, H-25, H-29), 0.90 (3H, m, CH_3_), 0.73 (3H, s, H-26), 0.67 (3H, s, H-24); ^13^C NMR (DMSO-d_6_, 100 MHz) *δ* 182.0 (C-28), 146.5 (C-13), 137.2 (Imid), 124.0 (Imid), 122.7 (Imid), 120.2 (C-12), 77.3 (C-3), 55.4 (C-5), 49.2 (CH_2_N), 47.8 (C-9), 47.4 (C-19), 46.1 (C-17), 42.2 (C-18), 41.9 (C-14), 39.2 (C-8), 38.8 (C-4), 38.6 (C-1), 37.1 (C-10), 36.2 (CH_3_N), 34.8 (C-21), 33.8 (C-29), 33.5 (C-7), 33.2 (C-22), 31.1 (C-20), 31.0 (CH_2_), 29.9 (CH_2_), 28.7 (C-23), 28.1 (C-15), 27.4 (C-2), 26.1 (C-27), 25.6 (CH_2_), 24.2 (C-30), 23.9 (C-16), 23.4 (C-11), 22.3 (CH_2_), 18.6 (C-6), 17.8 (C-26), 16.5 (C-24), 15.6 (C-25), 14.3 (CH_3_); anal. calcd for C_40_H_66_N_2_O_3_: C, 77.12; H, 10.68; found C, 76.96; H, 10.65.

##### 1-Hexyl-3-methylimidazolium (3*β*)-3-hydroxyolean-12-en-28-oylate (**15d**) [BnMIM][Olean]

Yield: 0.60 g, 96%, white solid, mp 228–230 °C. [α]_D_^18^ + 37.0 (c 0.84, MeOH); IR (KBr) ν_max_ 2943, 2859, 1652, 1547, 1388, 1162, 1086, 1047, 997, 720, 623 cm^−1^; ^1^H NMR (DMSO-d_6_, 400 MHz) *δ* 9.36 (1H, s, Imid), 7.81 (1H, m, Imid), 7.73 (1H, m, Imid), 7.44–7.38 (5H, m, Ph), 5.44 (2H, s, CH_2_Ph), 5.01 (1H, m, H-12), 4.30 (1H, m, OH),3.87 (3H, s, NCH_3_), 2.99 (1H, m, H-3), 2.90 (1H, d, ^2^*J* = 10.4 Hz, H-18), 1.94–0.642 (22H, m), 1.05 (3H, s, H-27), 0.89 (3H, s, H-23), 0.87 (3H, s, H-30), 0.84 (6H, s, H-25, H-29), 0.73 (3H, s, H-26), 0.68 (3H, s, H-24); ^13^C NMR (DMSO-d_6_, 100 MHz) *δ* 181.8 (C-28), 146.7 (C-13), 137.4 (Imid), 135.5 (Ph), 129.4 (Ph, signals of 2C), 129.2 (Ph), 128.8 (Ph, signals of 2C), 124.4 (Imid), 122.8 (Imid), 120.0 (C-12), 77.3 (C-3), 55.4 (C-5), 52.2 (CH_2_Ph), 47.8 (C-9), 47.5 (C-19), 46.0 (C-17), 42.3 (C-18), 41.9 (C-14), 39.3 (C-8), 38.8 (C-4), 38.6 (C-1), 37.1 (C-10), 36.3 (CH_3_N), 34.8 (C-21), 33.9 (C-29), 33.5 (C-7), 33.2 (C-22), 31.2 (C-20), 28.7 (C-23), 28.2 (C-15), 27.5 (C-2), 26.1 (C-27), 24.2 (C-30), 23.9 (C-16), 23.4 (C-11), 18.6 (C-6), 17.8 (C-26), 16.5 (C-24), 15.6 (C-25); anal. calcd for C_41_H_58_N_2_O_3_: C, 78.30; H, 9.62; found C, 78.16; H, 9.59.

##### 1-Ethyl-3-methylimidazolium (3*β*)-3-hydroxyours-12-en-28-oylate (**16a**) [EMIM][Urs]

Yield: 0.54 g, 95%, white solid, mp 218–220 °C. [α]_D_^18^ + 30.3 (c 0.89, MeOH); IR (KBr) ν_max_ 2925, 2852, 1639, 1554, 1461, 1385, 1305, 1288, 1169, 1106, 1045, 1030, 999, 723, 619 cm^−1^; ^1^H NMR (DMSO-d_6_, 400 MHz) *δ* 9.34 (1H, s, Imid), 7.84 (1H, m, Imid), 7.75 (1H, m, Imid), 5.01 (1H, m, H-12), 4.21 (2H, q, *J* = 7.2 Hz, CH2N), 3.84 (3H, s, NCH_3_), 3.00 (1H, br t, *J* = 6.0 Hz, H-3), 2.22 (1H, d, ^2^*J* = 11.6 Hz, H-18), 2.06–0.64 (22H, m), 1.42 (3H, t, *J* = 7.2 Hz, CH_3_), 1.01 (3H, s, H-27), 0.89 (6H, br s, H-23, H-30), 0.87 (3H, s, H-25), 0.81 (3H, d, *J* = 6.4 Hz, H-29), 0.78 (3H, s, H-24), 0.68 (3H, s, H-26); ^13^C NMR (DMSO-d_6_, 100 MHz) *δ* 181.5 (C-28), 140.5 (C-13), 136.9 (Imid), 124.0 (Imid), 123.3 (C-12), 122.4 (Imid), 77.3 (C-3), 55.4 (C-5), 53.8 (C-18), 47.8 (C-9), 47.4 (C-17), 44.6 (CH_2_N), 42.3 (C-14), 39.5 (C-8), 39.4 (C-19), 38.8 (C-1, C-4, C-20), 37.7 (C-22), 37.1 (C-10), 36.2 (CH_3_N), 33.5 (C-7), 31.6 (C-21), 28.8 (C-23), 28.6 (C-15), 27.5 (C-2), 25.2 (C-16), 23.8 (C-27), 23.4 (C-11), 21.9 (C-30), 18.6 (C-6), 17.9 (C-26, C-29), 16.6 (C-24), 15.7 (C-25), 15.6 (CH_3_); anal. calcd for C_36_H_58_N_2_O_3_: C, 76.28; H, 10.31; found C, 76.10; H, 10.29. MALDI TOF: *m*/*z* 478.472 (calculated for anion C_30_H_47_O_3_ [M + Na]^+^, calcd 478.342).

##### 1-Butyl-3-methylimidazolium (3*β*)-3-hydroxyours-12-en-28-oylate (**16b**) [BMIM][Urs]

Yield: 0.57 g, 96%, white solid, mp 212–214 °C. [α]_D_^24^ + 32.2 (c 0.87, MeOH); IR (KBr) ν_max_ 2924, 2852, 1685, 1647, 1541, 1461, 1385, 1307, 1168, 1078, 1044, 1031, 997, 973, 723, 624 cm^−1^; ^1^H NMR (DMSO-d_6_, 400 MHz) *δ* 9.35 (1H, s, Imid), 7.82 (1H, m, Imid), 7.75 (1H, m, Imid), 5.00 (1H, m, H-12), 4.19 (2H, t, *J* = 6.8 Hz, CH_2_N), 3.87 (3H, s, NCH_3_), 2.99 (1H, m, H-3), 2.22 (1H, d, ^2^*J* = 11.2 Hz, H-18), 2.05–0.62 (22H, m), 1.77 (2H, m, CH_2_), 1.25 (2H, m, CH2), 0.99 (3H, s, H-27), 0.89 (9H, br s, H-23, H-30, CH_3_), 0.86 (3H, s, H-25), 0.79 (3H, d, *J* = 6.0 Hz, H-29), 0.76 (3H, s, H-24), 0.67 (3H, s, H-26); ^13^C NMR (DMSO-d_6_, 100 MHz) *δ* 181.7 (C-28), 140.5 (C-13), 137.2 (Imid), 124.0 (Imid), 123.2 (C-12), 122.7 (Imid), 77.3 (C-3), 55.4 (C-5), 53.8 (C-18), 48.9 (CH_2_N), 47.8 (C-9), 47.5 (C-17), 42.3 (C-14), 39.5 (C-8), 39.4 (C-19), 38.8 (C-1, C-4, C-20), 37.7 (C-22), 37.1 (C-10), 36.2 (CH_3_N), 33.5 (C-7), 31.9 (CH_2_), 31.7 (C-21), 28.8 (C-23), 28.6 (C-15), 27.5 (C-2), 25.2 (C-16), 23.8 (C-27), 23.4 (C-11), 21.9 (C-30), 19.2 (CH_2_), 18.6 (C-6), 17.9 (C-26, C-29), 16.5 (C-24), 15.7 (C-25), 13.7 (CH_3_); anal. calcd for C_38_H_62_N_2_O_3_: C, 76.72; H, 10.52; found C, 76.59; H, 10.49.

##### 1-Hexyl-3-methylimidazolium (3*β*)-3-hydroxyours-12-en-28-oylate (**16c**) [HMIM][Urs]

Yield: 0.59 g, 96%, white solid, mp 190–192 °C. [α]_D_^22^ + 32.3 (c 0.88, MeOH); IR (KBr) ν_max_ 2925, 2856, 1647, 1619,1567, 1455, 1385, 1243, 1168, 1105, 1045, 1029, 974, 753, 622 cm^−1^; ^1^H NMR (DMSO-d_6_, 400 MHz) *δ* 9.39 (1H, s, Imid), 7.81 (1H, m, Imid), 7.74 (1H, m, Imid), 4.99 (1H, m, H-12), 4.18 (2H, t, *J* = 7.2 Hz, CH_2_N), 3.87 (3H, s, NCH_3_), 2.99 (1H, br t, *J* = 7.2 Hz, H-3), 2.22 (1H, d, ^2^*J* = 11.2 Hz, H-18), 2.05–0.62 (22H, m), 1.77 (2H, m, CH_2_), 1.26 (6H, m, CH_2_), 0.99 (3H, s, H-27), 0.89 (9H, br s, H-23, H-30, CH_3_), 0.86 (3H, s, H-25), 0.79 (3H, d, *J* = 6.4 Hz, H-29), 0.76 (3H, s, H-24), 0.67 (3H, s, H-26); ^13^C NMR (DMSO-d_6_, 100 MHz) *δ* 181.5 (C-28), 140.5 (C-13), 137.3 (Imid), 124.0 (Imid), 123.2 (C-12), 122.7 (Imid), 77.3 (C-3), 55.4 (C-5), 53.8 (C-18), 49.2 (CH_2_N), 47.8 (C-9), 47.4 (C-17), 42.2 (C-14), 39.5 (C-8), 39.4 (C-19), 38.8 (C-1, C-4, C-20), 37.7 (C-22), 37.0 (C-10), 36.2 (CH_3_N), 33.5 (C-7), 31.9 (CH_2_), 31.7 (C-21), 31.0 (CH_2_), 28.8 (C-23), 28.6 (C-15), 27.5 (C-2), 25.6 (CH_2_), 25.2 (C-16), 23.8 (C-27), 23.3 (C-11), 22.4 (CH_2_), 21.9 (C-30), 18.6 (C-6), 17.9 (C-26, C-29), 16.6 (C-24), 15.7 (C-25), 14.3 (CH_3_); anal. calcd for C_40_H_66_N_2_O_3_: C, 77.12; H, 10.68; found C, 76.96; H, 10.65.

##### 1-Benzyl-3-methylimidazolium (3*β*)-3-hydroxyours-12-en-28-oylate (**16d**) [BnMIM][Urs]

Yield: 0.59 g, 94%, white solid, mp 207–209 °C. [α]_D_^22^ + 26.9 (c 0.80, MeOH); IR (KBr) ν_max_ 2923, 2868, 1633, 1556, 1455, 1386, 1288, 1161, 1090, 1045, 1029, 917, 721, 622 cm^−1^; ^1^H NMR (DMSO-d_6_, 400 MHz) *δ* 9.51 (1H, s, Imid), 7.85 (1H, m, Imid), 7.76 (1H, m, Imid), 7.48–7.37 (5H, m, Ph), 5.48 (2H, s, CH_2_Ph), 5.00 (1H, m, H-12), 3.88 (3H, s, NCH_3_), 2.99 (1H, br t, *J* = 6.8 Hz, H-3), 2.22 (1H, d, ^2^*J* = 11.2 Hz, H-18), 2.05–0.62 (22H, m), 0.99 (3H, s, H-27), 0.89 (6H, br s, H-23, H-30), 0.85 (3H, s, H-25), 0.79 (3H, d, *J* = 6.0 Hz, H-29), 0.75 (3H, s, H-24), 0.67 (3H, s, H-26); ^13^C NMR (DMSO-d_6_, 100 MHz) *δ* 181.3 (C-28), 140.4 (C-13), 137.4 (Imid), 135.5 (Ph), 129.4 (Ph, signals of 2C), 129.1 (Ph), 128.8 (Ph, signals of 2C), 124.4 (Imid), 123.4 (C-12), 122.8 (Imid), 77.3 (C-3), 55.4 (C-5), 53.8 (C-18), 52.2 (CH_2_Ph), 47.7 (C-9), 47.4 (C-17), 42.2 (C-14), 39.5 (C-8), 39.4 (C-19), 38.8 (C-1, C-4, C-20), 37.6 (C-22), 37.0 (C-10), 36.3 (CH_3_N), 33.5 (C-7), 31.6 (C-21), 28.8 (C-23), 28.5 (C-15), 27.5 (C-2), 25.1 (C-16), 23.8 (C-27), 23.3 (C-11), 21.9 (C-30), 18.6 (C-6), 17.9 (C-26, C-29), 16.6 (C-24), 15.7 (C-25); anal. calcd for C_41_H_60_N_2_O_3_: C, 78.30; H, 9.62; found C, 78.16; H, 9.58.

##### 1-Ethyl-3-methylimidazolium (3*β*)-3-hydroxyolup-20(29)-en28-oylate (**17a**) [EMIM][Lup]

Yield: 0.54 g, 96%, white solid, mp 253–255 °C. [α]_D_^24^ + 1.9 (c 0.85, MeOH); IR (KBr) ν_max_ 2925, 2855, 1639, 1557, 1458, 1377, 1302, 1257, 1208, 1182, 1083, 1034, 1007, 923, 722, 623 cm^−1^; ^1^H NMR (DMSO-d_6_, 400 MHz) *δ* 9.33 (1H, s, Imid), 7.83 (1H, m, Imid), 7.74 (1H, m, Imid), 4.61 (1H, br s, H-29), 4.48 (1H, br s, H-29), 4.21 (2H, q, *J* = 7.2 Hz, CH_2_N), 3.87 (3H, s, CH_3_N), 3.18 (1H, m, H-19), 2.97 (1H, m, H-3), 2.61 (1H, m, H-13), 2.20–0.60 (23H, m), 1.61 (3H, s, H-30), 1.42 (3H, t, *J* = 7.6 Hz, CH_3_), 0.88 (6H, s, H-23, H-27), 0.86 (3H, s, H-26), 0.76 (3H, s, H-25), 0.62 (3H, s, H-24); ^13^C NMR (DMSO-d_6_, 100 MHz) *δ* 180.6 (C-28), 152.5 (C-20), 136.9 (Imid), 124.0 (Imid), 122.4 (Imid), 108.9 (C-29), 77.2 (C-3), 56.6 (C-17), 55.5 (C-5), 50.7 (C-9), 49.7 (C-18), 47.4 (C-19), 44.6 (CH_2_N), 42.4 (C-14), 40.8 (C-8), 38.9 (C-4), 38.8 (C-1), 38.3 (C-22), 37.7 (C-13), 37.2 (C-10), 36.2 (CH_3_N), 34.6 (C-7), 34.1 (C-16), 31.3 (C-21), 30.0 (C-15), 28.6 (C-23), 27.6 (C-2), 25.9 (C-12), 21.2 (C-11), 19.6 (C-30), 18.5 (C-6), 16.7 (C-25), 16.5 (C-24), 16.3 (C-26), 15.6 (CH_3_), 14.8 (C-27); anal. calcd for C_36_H_58_N_2_O_3_: C, 76.28; H, 10.31; found C, 76.09; H, 10.28. MALDI TOF: *m*/*z* 478.542 (calculated for anion C_30_H_47_O_3_ [M + Na]^+^, calcd 478.342).

##### 1-Butyl-3-methylimidazolium (3*β*)-3-hydroxyolup-20(29)-en28-oylate (**17b**) [BMIM][Lup]

Yield: 0.58 g, 97%, white solid, mp 160–162 °C. [α]_D_^24^ + 2.4 (c 0.75, MeOH); IR (KBr) ν_max_ 2928, 2863, 1638, 1564, 1449, 1371, 1257, 1205, 1183, 1034, 1007, 878, 753, 623 cm^−1^; ^1^H NMR (DMSO-d_6_, 400 MHz) *δ* 9.42 (1H, s, Imid), 7.82 (1H, m, Imid), 7.75 (1H, m, Imid), 4.61 (1H, br s, H-29), 4.47 (1H, br s, H-29), 4.18 (2H, t, *J* = 7.2 Hz, CH_2_N), 3.88 (3H, s, CH_3_N), 3.18 (1H, m, H-19), 2.97 (1H, m, H-3), 2.62 (1H, m, H-19), 2.20–0.60 (23H, m), 1.77 (2H, m, CH_2_), 1.61 (3H, s, H-30), 1.25 (2H, m, CH_2_), 0.90 (3H, m, CH_3_), 0.88 (6H, s, H-23, H-27), 0.86 (3H, s, H-26), 0.76 (3H, s, H-25), 0.64 (3H, s, H-24); ^13^C NMR (DMSO-d_6_, 100 MHz) *δ* 180.4 (C-28), 152.6 (C-20), 137.2 (Imid), 124.1 (Imid), 122.7 (Imid), 108.9 (C-29), 77.2 (C-3), 56.6 (C-17), 55.5 (C-5), 50.7 (C-9), 49.7 (C-18), 48.9 (CH_2_N), 47.4 (C-19), 42.5 (C-14), 40.8 (C-8), 38.9 (C-4), 38.8 (C-1), 38.4 (C-22), 37.7 (C-13), 37.2 (C-10), 36.2 (CH_3_N), 34.7 (C-7), 34.1 (C-16), 31.9 (CH_2_), 31.3 (C-21), 30.0 (C-15), 28.6 (C-23), 27.7 (C-2), 25.9 (C-12), 21.2 (C-11), 19.6 (C-30), 19.2 (CH_2_), 18.5 (C-6), 16.7 (C-25), 16.5 (C-24), 16.3 (C-26), 14.8 (C-27), 13.7 (CH_3_); anal. calcd for C_38_H_62_N_2_O_3_: C, 76.72; H, 10.50; found C, 76.57; H, 10.48.

##### 1-Hexyl-3-methylimidazolium (3*β*)-3-hydroxyolup-20(29)-en28-oylate (**17c**) [HMIM][Lup]

Yield: 0.60 g, 97%, white solid, mp 168–170 °C. [α]_D_^22^ + 4.2 (c 0.69, MeOH); IR (KBr) ν_max_ 2928, 2861, 1638, 1561, 1449, 1374, 1257, 1208, 1180, 1036, 1007, 878, 754, 623 cm^−1^; ^1^H NMR (DMSO-d_6_, 400 MHz) *δ* 9.42 (1H, s, Imid), 7.82 (1H, m, Imid), 7.75 (1H, m, Imid), 4.61 (1H, br s, H-29), 4.47 (1H, br s, H-29), 4.18 (2H, t, *J* = 7.2 Hz, CH_2_N), 3.88 (3H, s, CH_3_N), 3.18 (1H, m, H-19), 2.97 (1H, br t, *J* = 7.6 Hz, H-3), 2.62 (1H, m, H-19), 2.20–0.60 (23H, m), 1.77 (2H, m, CH_2_), 1.61 (3H, s, H-30), 1.26 (6H, m, CH_2_), 0.89 (3H, m, CH_3_), 0.88 (6H, s, H-23, H-27), 0.86 (3H, s, H-26), 0.76 (3H, s, H-25), 0.62 (3H, s, H-24); ^13^C NMR (DMSO-d_6_, 100 MHz) *δ* 180.4 (C-28), 152.5 (C-20), 137.3 (Imid), 124.0 (Imid), 122.7 (Imid), 108.9 (C-29), 77.2 (C-3), 56.6 (C-17), 55.5 (C-5), 50.7 (C-9), 49.7 (C-18), 49.2 (CH_2_N), 47.4 (C-19), 42.5 (C-14), 40.8 (C-8), 38.9 (C-4), 38.8 (C-1), 38.4 (C-22), 37.7 (C-13), 37.2 (C-10), 36.2 (CH_3_N), 34.7 (C-7), 34.1 (C-16), 31.3 (C-21), 31.0 (CH_2_), 30.0 (C-15), 29.9 (CH_2_), 28.6 (C-23), 27.7 (C-2), 25.9 (C-12), 25.6 (CH_2_), 22.4 (CH_2_), 21.2 (C-11), 19.6 (C-30), 18.5 (C-6), 16.7 (C-25), 16.5 (C-24), 16.3 (C-26), 14.8 (C-27), 14.3 (CH_3_); anal. calcd for C_40_H_66_N_2_O_3_: C, 76.28; H, 10.31; found C, 76.09; H, 10.28.

##### 1-Benzyl-3-methylimidazolium (3*β*)-3-hydroxyolup-20(29)-en28-oylate (**17d**) [BnMIM][Lup]

Yield: 0.60 g, 96%, white solid, mp 181–183 °C. [α]_D_^24^ + 2.1 (c 0.76, MeOH); IR (KBr) ν_max_ 2926, 2859, 1652, 1567, 1446, 1373, 1255, 1207, 1166, 1035, 1007, 877, 752, 620 cm^−1^; ^1^H NMR (DMSO-d_6_, 400 MHz) *δ* 9.49 (1H, s, Imid), 7.84 (1H, m, Imid), 7.75 (1H, m, Imid), 7.47–7.37 (5H, m, Ph), 5.47 (2H, s, CH_2_Ph), 4.61 (1H, br s, H-29), 4.48 (1H, br s, H-29), 3.88 (3H, s, CH_3_N), 3.18 (1H, m, H-19), 2.98 (1H, br t, *J* = 7.6 Hz, H-3), 2.62 (1H, m, H-19), 2.20–0.60 (23H, m), 1.61 (3H, s, H-30), 0.89 (6H, s, H-23, H-27), 0.87 (3H, s, H-26), 0.76 (3H, s, H-25), 0.62 (3H, s, H-24); ^13^C NMR (DMSO-d_6_, 100 MHz) *δ* 180.4 (C-28), 152.5 (C-20), 137.3 (Imid), 135.5 (Ph), 129.4 (Ph, signals of 2C), 129.2 (Ph), 128.8 (Ph, signals of 2C), 124.4 (Imid), 122.8 (Imid), 108.9 (C-29), 77.3 (C-3), 56.6 (C-17), 55.5 (C-5), 52.2 (CH_2_Ph), 50.7 (C-9), 49.7 (C-18), 47.4 (C-19), 42.5 (C-14), 40.8 (C-8), 38.9 (C-4), 38.8 (C-1), 38.4 (C-22), 37.7 (C-13), 37.2 (C-10), 36.3 (CH_3_N), 34.7 (C-7), 34.1 (C-16), 31.3 (C-21), 30.0 (C-15), 28.6 (C-23), 27.7 (C-2), 25.9 (C-12), 21.2 (C-11), 19.6 (C-30), 18.5 (C-6), 16.7 (C-25), 16.5 (C-24), 16.3 (C-26), 14.8 (C-27); anal. calcd for C_41_H_60_N_2_O_3_: C, 78.30; H, 9.62; found C, 78.06; H, 9.58.

## 3. Results

### 3.1. Chemistry

In order to implement the planned studies, our group started with developing a strategy for the synthesis of imidazole derivatives linked to the triterpenoid molecules (betulinic, ursolic and oleanolic acids) for the subsequent preparation of ionic compounds based on them.

At first, imidazole derivatives of oleanolic acid **3**, **4** (Figure 1) were synthesized, which were supposed to be used as reference substances with their structurally similar ionic analogs as well. The target compounds were synthesized in three stages by the addition of hexanediol to the oleanolic acid molecule, through the intermediate preparation of its acid chloride, and by the esterification of the resulting alcohol **2** with carboxylic acids containing an imidazole fragment at the final stage.

The synthesis of triterpenoids covalently linked to the *N*-methylimidazole cation **9–11**, synthesized by the successive addition reactions of α,ω-dibromoalkanes (1,2-dibromoethane, 1,4-dibromobutane, 1,6-dibromohexane) to triterpenoids (oleanolic, ursolic and betulinic acids) and the addition of *N*-methylimidazole to the resulting bromides to obtain target ionic compounds with different numbers of methylene units separating the triterpenoid and imidazole molecule was performed according to Figure 2.

The ionic compounds **12a**–**12f** were synthesized similarly to the above approach to the preparation of imidazole derivatives based on oleanolic acid and a number of nitrogen-containing compounds: pyridine, piperidine, morpholine, pyrrolidine, triethylamine and dimethylethanolamine, as shown in Figure 3.

Based on imidazole bromide of oleanolic acid **9c**, imidazole derivatives of oleanolic acid with different anions **13a**–**13g** were synthesized for the first time as well (Figure 4).

Furthermore, we synthesized ionic compounds based on imidazole bromide, with the triterpenoids (oleanolic, ursolic and betulinic acids) used as anions (Figure 5).

### 3.2. Biological Evaluation

#### Cytotoxic Activity In Vitro

The antitumor activity of the synthesized ionic compounds was studied in vitro in a number of tumor cell lines.

The antitumor activity of the synthesized ionic compounds was initially assessed in vitro on Jurkat, K562, U937, HL60, HEK293, A2780 cell lines, including the determination of IC50 by flow cytometry involving Guava ViaCount (Millipore) reagent kits.

The results prove that introduction of an imidazole fragment into the oleanolic acid molecule reduces its cytotoxic activity (compounds **3** and **4**, Table 1). The derivatives of oleanolic and ursolic acids **9d** and **10c** exhibited the highest activity among the ionic compounds based on triterpenoids covalently linked to the *N*-methylimidazole cation, with the number of methylene units separating the molecule of triterpenoid and imidazole equal to six. The least activity was shown by hydroxy derivatives of oleanolic acid **9b** and **9c** with the number of methylene units n = 4 and 6.

The replacement of the imidazole fragment by a pyridine one also contributed to an increase in the cytotoxic activity of the ionic compounds based on oleanolic acid; the activity of other derivatives was shown to decrease in the series triethylamine > piperidine > pyrrolidine > dimethylethanolamine > morpholine.

The substitution of bromine for other counterions in some cases leads to a slight increase in the activity of the ionic compound, for example, in the case of BF_6_-, PF_6_- and CH_3_COO-, while in other cases there was a slight decrease in the cytotoxicity of SbF_6_-, C_6_H_5_SO_3_-, *m*-C_6_H_4_(OH)COO- and CH_3_CH(OH)COO-.

Ionic imidazole compounds with triterpenoids as anions do not exhibit significant cytotoxic activity against the selected cell lines.

At the next stage, we studied the mechanism of action of ionic compounds–derivatives of pentacyclic triterpenoids in cells. We analyzed the processes of damage to mitochondria by ionic compounds depending on the damage to the mitochondrial membrane and the ability to initiate apoptosis according to the mitochondrial type. This mechanism of action of ionic compounds was detailed in our previous work on ionic compounds with simpler structures [17].

The cells with apoptosis initiated due to damage to the mitochondrial membrane exhibit depolarization of the mitochondrial electrochemical gradient, release of apoptogenic molecules and activation of caspases, loss of plasma membrane symmetry, changes in chromatin and, as a result, rupture of the plasma membrane [18]. Multiparametric evaluation of marker molecules, along with the detection of damage to the mitochondrial membrane, makes it possible to clearly determine the type of cell death. The method of cell staining with three different dyes (Annexin V Alexa Fluor488, 7AAD and MitoSense Red) provides simultaneous specification of various aspects of the state of the cell membrane, the state of DNA and the viability of mitochondria; together, it is a reliable sign of intrinsic and mitochondrial apoptosis pathway. Normal cells with intact mitochondrial membranes have high MitoSense Red fluorescence, while cells with decoupled oxidation and phosphorylation have much lower fluorescence.

The control cells without apoptotic events were found to contain phosphatidylserine on the inner surface of the membrane, while apoptotic cells exhibit a positive green fluorescence due to the externalization of phosphatidylserine, which has a high affinity for annexin. The DNA intercalator 7-aminoactinomycin (7-AAD), which does not penetrate into a living cell, appears inside the cell only at the late stages of apoptosis when the membrane is damaged, usually observed in late apoptosis and cell necrosis. ICs with a long side alkyl radical have the greatest damaging effect on cells [4,19,20,21]. To date, a great deal of varying data on the mechanism of action of ionic compounds has been reported. Some researchers claim that ICs are inducers of apoptosis, damage the cell cycle, have genotoxicity and cause oxidative stress in cells [19,22,23,24]. Other researchers believe that ionic compounds act on the cell rather rapidly, without causing programmed cell death, but initiating rapid necrosis due to a sharp shift in the ion gradient and destruction of the cell membrane [25]. When studying the mode of death of keratinocytes under the use of biocides based on ionic compounds, applying cell sorting with fluorescence activation (FACS) and parallel measurement of caspase 3/7 activation, necrosis was found to be the leading cytotoxic mechanism at high concentrations of biocides containing [C14MIM]Cl and [C14quin] [26]. Moreover, apoptosis was observed precisely at borderline IC50 values. It is noteworthy that the modification of the anion had a significant effect on cytotoxicity in this research. The use of N[SO_2_CF_3_] as an anion for [C16MIM] attenuated the cytotoxicity of the ionic compound by a factor of 10 compared to other anions, thereby confirming the fact that cytotoxicity can be a controlled property when using ionic compounds as biocides [26]. In 2018, exposure to the ionic compound [C8mim]Br was found to alter levels of heat shock protein 70 (HSP70) and HSP90, in general suppressing total antioxidant capacity, depleting heme oxygenase-1 (HO-1) and increasing transcription and inducible nitric oxide synthase (iNOS) activity in HepG2 cells. These results indicate that [C8mim]Br causes significant biochemical damage and induces oxidative stress in HepG2 cells. Furthermore, increased p53 phosphorylation, mitochondrial membrane disruption, cyclooxygenase-2 activation, modulation of Bcl-2 family proteins, release of cytochrome c and Smac/DIABLO complex and inhibition of apoptosis-inhibiting protein-2 (c-IAP2) and survivin observed in the cells treated with [C8mim]Br suggest that [C8mim]Br-induced apoptosis may be mediated by the mitochondrial pathway. Further studies have shown that [C8mim]Br exposure increases transcription and tumor necrosis factor *α* (TNF-*α*) content and promotes Fas and FasL expression, indicating that TNF-*α* and Fas/FasL are involved in [C8mim]Br-induced apoptosis. Moreover, [C8mim]Br cytotoxicity was partially inhibited by *N*-acetylcysteine (NAC), and NAC reversed [C8mim]Br-mediated mitochondrial dysfunction and blocked apoptotic events by inducing glutathione production, thereby inhibiting reactive oxygen species (ROS- ions) [27]. This work demonstrates that the ROS-mediated mitochondrial and death receptor mediated apoptosis pathway is involved in [C8mim]Br-induced cell apoptosis, and reveals a major issue of genotoxicity of ionic compounds. If they are genotoxic, what types of DNA damage are caused by ionic compounds and how this feature of such molecules can be used in the development of antitumor compounds are issues that we will try to address in our work.

In 2019, Silva et al. synthesized a line of ionic compounds from betulinic acid that exhibited antitumor activity against RKO, T47D, MG63, A549, HepG2 cell lines and normal HFF-1 fibroblasts by involving MEK, NF-kB and JNK signaling pathways, which are significant for oncogenesis [28]. Earlier, a high antiviral potential of ionic compounds based on betulinic acid was reported with several modes of antiviral action addressed in the work: inhibition of HIV protease (asportyl protease), impaired virion assembly in the cytoplasm and inhibition of reverse transcriptase [29]. The concept of dual biological function of an ionic compound, reported in some works on the mechanisms of action in the cell, has recently become more widespread [30]. For example, sodium ibuprofen (anti-inflammatory agent) can be combined with didecyldimethylammonium bromide (antibacterial and anti-inflammatory effect) to form a complex ionic compound—didecyldimethylammonium ibuprofen retaining a dual biological function due to the presence of an ion and a counterion [31]. This synergy of action of ions and counterions in one molecule in various pentacyclic triterpenoids apparently provides better aqueous solubility and significant improvement of antitumor and antiviral properties of the molecule, including an additional biological function introduced through the counterion as well.

The study of the line of synthesized ionic compounds of triterpenoid derivatives against mitochondria of Jurkat tumor cells revealed that oleanolic acid derivatives exhibited the highest activity in relation to the mitochondrial membrane potential that induce apoptosis due to damaged mitochondria (Figure 1). The highest percentage of damaged mitochondria was observed in compounds **12a**, **13d**, **13g**, **12e**, **12f**, **12c** and **12d** (97.89%, 97.47%, 97.4%, 97.3%, 86.98%, 74.26% and 62.11%, respectively). The same high potential is registered in compounds **12a**, **13d**, **12e** and **12f** concerning the percentage of cells in the apoptotic state, namely 87.81%, 98.32%, 83.25% and 85.10%, respectively, while two compounds **13g** and **12c** show a fairly low content of apoptotic cells (25% and 35%, respectively) along with a high content of cells with damaged mitochondria (97.4% and 74.26%, respectively). Oleanolic acid derivatives seem to have higher water solubility, but they have a high lipophilicity in the ionic form like almost all pentacyclic structures, and this fact may account for their better bioavailability and cell membrane permeability. In terms of the effect on mitochondria, the above derivatives of oleanolic acid are comparable to staurosporine, considered a reference molecule in almost all studies due to its ability to induce both caspase-dependent and caspase-independent forms of apoptosis [32].

To demonstrate the mitochondrial pathway of apoptosis in tumor cells, the process of cytochrome c release from mitochondria under the influence of derivatives of oleanolic, ursolic and betulinic acids was studied. Staurosporine (an inhibitor of most kinases in the cell) was taken as a control compound. Viable or living cells exhibit higher levels of cytochrome c fluorescence, while the initiation of the apoptosis process in the cells that have released cytochrome c from mitochondria results in low levels of fluorescence observed in the cytoplasm when using anti-cytochrome C antibody stained with FITC.

For example, the cytofluorimetry analysis of Jurkat tumor cells treated with compounds **13d** and **12a** showed the following values of cytochrome c release from mitochondria: 19.43% for compound **13d** and 40.58% for compound **12a** (Figure 2c,d).

According to the study of the cell cycle applying Guava Cell Cycle Reagent, ionic compounds based on oleanolic acid (**12a**, **13d** and **13f**) caused the most pronounced decrease in the G2 peak, almost on a par with cisplatin (Figure 3), while the S phase was reduced to a lesser extent. Our findings are consistent with the literature data that many ICs predominantly affect the G2 and S phases, reducing them [33,34,35,36]. Compounds **13d** and **13f** also turned out to be inducers of the hypodiploid cell population (sub-G1 phase—6.91% and 9.83%) in Jurkat tumor cells under appropriate treatment with the test substances. Indeed, the hypodiploid cell population at similar concentrations for Jurkat tumor line was higher for cisplatin (32.27%) compared with the control (1.61%). Therefore, the results suggest that ICs, as well as cisplatin, are inducers of apoptosis in various cell lines under study, probably due to DNA damage, as indicated by the appearance of a hypodiploid cell population within 24 h [23,37].

Thus, ionic compounds proved to have a universal mechanism of action. First, they penetrate through the mitochondrial membrane, disrupt functions of the respiratory chain, promote cytochrome release into the cytoplasm and initiate mitochondrial pathway of apoptosis by activating the caspase cascade. Secondly, they activate apoptosis by a p53-dependent mechanism by damaging DNA, confirmed by the study of the cell cycle. To show the effect of ionic compounds on DNA, as well as to address the issue of genotoxicity of ionic compounds based on pentacyclic triterpenoids, we analyzed the most universal intracellular signaling pathways responsible for cell proliferation and initiation of apoptosis under treatment with compounds **13d**, **12a** and **12e** by the Luminex xMAP technology (Figure 4 and Figure 5). These compounds were selected on the basis of differences in their ability to initiate mitochondrial apoptosis, dissipate oxidation and phosphorylation processes and induce necrosis in cells (Table 1).

The determination of proteins in a cell by Luminex xMAP technology is characterized by the presence of multiplex multiparametric analysis with the simultaneous use of a multiplex of antibodies with a fluorescent label and magnetic particles, thereby resulting in an increase in the sensitivity of the technique and expansion of the dynamic range, as well as an increase in detection accuracy. Luminex XMAP technology provides a simultaneous analysis of a sufficient number of different proteins in various types of biological fluids (plasma, serum, cerebrospinal fluid, follicular fluid, tissue lysate, etc.) and in cell supernatants. Consequently, multiplexing saves time/costs and ensures the same conditions for the measurements, which is essential for significance and statistical processing. Simultaneous measurement of cytokines by bead multiplex analysis has certain similarities with ELISA, in particular the use of antibodies, but has a fundamental difference in using magnetic fluorescent beads coupled to antikinase monoclonal antibodies.

Magnetic microspheres (stained inside with two fluorescent dyes) in combination with monoclonal antibodies against the analyzed analytes (the main intracellular kinases in this case) are incubated with the samples and standards; then, the samples/standards are incubated with biotinylated monoclonal antibodies against the analyzed analytes and, finally, with a solution of streptavidin–phycoerythrin. The Luminex technology identifies the different proteins present in each well and converts the mean fluorescence intensity (MFI) of each measured cytokine to pg/mL using software and standard curves. This approach is successfully applied in fundamental and clinical research. In this work, we analyzed various protein analytes that make up the phosphorylated and non-phosphorylated fractions of the main kinases in the signaling pathways: CREB, JNK, NFkB, p38, ERK1/2, Akt, p70S6K, STAT3, STAT5 proteins in six samples of lysates of Jurkat tumor cells by Luminex technology (MILLIPLEX^®^ MAP 9-Plex Multi-Pathway 9-plex Magnetic Bead Kit). Additionally, the processes of apoptosis initiation in the cells and the genotoxicity of compounds **13d** and **12a** were analyzed applying MILLIPLEX^®^ MAP 7-plex DNA Damage/Genotoxicity Magnetic Bead Kit and MILLIPLEX^®^ MAP Early apoptosis 7-plex Magnetic Bead Kit panels detecting phosphorylated Chk1, Chk2, H2A.X, p53 kinases and total levels of ATR and p21 genome repair proteins.

In summary, a pairwise comparison of the phosphorylated and non-phosphorylated kinase fractions reveals that the main changes in the ratio of these two fractions are expressed for the Akt, p70S6K, ERK1\2, NfkB, CREB, STAT3 and STAT5 signaling pathways (Figure 4). RTK (receptor tyrosine kinase) signaling often causes an increase in metabolism and cell growth. ERK/MAP kinases and Akt are two key families of Ser/Thr kinases activated via RTK signaling, causing an increase in the activity of p70S6 kinases, Msk1, STAT3 (Ser727) and CREB, which also increase the activation of other intermediates. Other signaling pathways induced by stress or death receptors cause the activation of p38, JNK and NF-κB. As shown in Figure 4, compounds **12a** and **13d** work in the same way and most pronouncedly affect the STAT5, STAT3, Akt and p38 kinases in the tumor cell at a concentration of 0.006 and 0.004 µM, which affects the number of phosphorylated forms of various kinases compared to the control. However, the effect of the studied compounds is dose-dependent. Thus, the compounds **12a** and **13d** work in the same way on the protein profile of the studied signaling pathways. Still, the effect of this class of compounds is ambiguous due to rather contradictory data on the activation of intracellular signaling in the cell. Moreover, the available data indicate that the compounds under study exhibit high antitumor potential and are of unequivocal interest as drug candidates.

STAT kinases are responsible for fundamental processes in the cell, such as viability, proliferation, differentiation and immune response [38]. Currently, STAT3, STAT5A and STAT5B are considered important effectors of various cell transformations. Impaired intracellular STAT3, STAT5A and STAT5B signaling has been reported in various solid tumors such as prostate, colon, glioma, head and neck tumors, melanoma, hemoblastosis and breast cancer [39,40]. Recently, gain-of-function mutations in STAT5B and STAT3 have been found in patients with various types of leukemia and lymphomas [41,42,43]. Localized primarily in the SH2 protein domain, these mutations provide steady and prolonged signaling and are associated with poor survival prognosis and more recurrent disease in patients. Overall, all these data confirm the important role of STAT3 and STAT5A/5B kinases as therapeutic targets in oncological diseases [44,45]. Furthermore, there are reported data on STAT3 and STAT5 as tumor suppressors and their regulatory function in relation to the antitumor effect of immune cells [46,47]. The study of childhood forms of acute myeloid leukemia (AML) revealed that the patients whose AML cells exhibited low STAT3 activation had a lower event-free survival compared to the patients with a stronger STAT3 response. When measuring STAT3, STAT5 and ERK1/2 proteins, 113 patients included in the trial exhibited low inducible STAT3 activity, which was independently associated with lower survival and more frequent relapses of the disease. A similar pattern is observed in STAT5: the patients with the lowest and highest response to therapy had a lower event-free survival compared to the patients with an intermediate STAT5 response. Gene expression profiles in the patients with low inducible STAT3/5 activation were compared with the patients with higher inducible STAT3/5 signaling applying existing RNA sequencing data in AML. Furthermore, the genes encoding hematopoietic factors and subunits of the mitochondrial respiratory chain were found to be overexpressed in the groups with a low STAT3/5 response, thus pointing to a key role of these kinases in the development of resistance to chemotherapy and on the prognosis and severity of the disease [48]. Still, the role of STAT3 and STAT5 kinase proteins, as well as their interaction in various tumor processes, are to be detailed to develop therapeutic strategies selectively blocking or activating STAT3 and/or STAT5 in hematological malignancies and other cancers.

The specific role of STAT3 and STAT5 was assessed by changes in the content of tyrosine-phosphorylated forms of these proteins (STAT3(ph) and STAT5(ph)). The treatment of Jurkat cells with appropriate concentrations of the studied compounds revealed differences in the activation profiles of STAT3 and STAT5, indicating that the signaling function of these proteins apparently depends on the compound used in the experiment. A significant (3-fold) and lasting (up to 12 h) increase in the content of both STAT5 and STAT3 (to a lesser extent) is observed in the Jurkat cells treated with compounds **12a** (IC8) and **13d** (IC5). An inhibitory assay involving pharmacological agents selectively inhibiting the activity of JAK kinases associated with the IC-2 receptor proved a crucial role of the JAK3/STAT5 signaling pathway in triggering and maintaining IC-2-regulated human PBL proliferation [49]. Consequently, the high proliferative status of lymphoid cell cultures was found to be associated with high constitutive activity of STAT3 and STAT5. The significant difference between elevated concentrations of STAT5 and STAT3 in the cells treated with the studied compounds can be a reaction to stress, or the studied compounds affecting the Jurkat tumor culture are probably mitogen-like compounds.

The ATM/ATR signaling pathway is responsible for maintaining stability and integrity of the genome in somatic cells. Its activation occurs in response to DNA damage, followed by the activation of signaling pathways responsible for cell cycle arrest (the so-called checkpoint control), repair, apoptosis and aging. The initiation of a cellular response to DNA damage begins with the activation of ATM and ATR kinases, which recognize DNA damage and activate cellular signaling responsible for DNA repair. The ATM and ATR kinases play one of the key roles in the phosphorylation of histone H2AX at Ser139 in the area of DNA breaks to form the so-called *γ*H2AX foci, which is essential for initiating the repair of DNA double-strand breaks. ATM activation is supposed to proceed both directly through interaction with damaged DNA and through interaction with protein members of repair complexes. For example, ATM, as part of the BRCA1-associated genome surveillance complex (BASC), is involved in the recognition and repair of aberrant DNA structures. ATR also binds to the components of the nucleosome remodeling and deacetylation (NRD) complex, such as chromodomain helicase DNA binding protein 4 (CHD4) and histone deacetylase 2 (HDAC2) [50]. ATR is activated in the presence of single-stranded DNA, which is an intermediate product in nucleotide excision repair and DNA damage repair by homologous recombination [51]. The results suggest a model in which multiple checkpoint protein complexes independently localize to the sites of DNA damage and interact to trigger the checkpoint-signaling cascade. The analysis of the ATR level in the cells treated with compounds **12a** and **13d** indicated a significant increase in the level of ATR kinase in response to the damaging effect of these compounds, while the protein levels during the first 6 h significantly exceed the levels of ATR kinase after 12 h of incubation (*p* > 0.005). A comparison of the effect of compounds **12a** and **13d** on cells reveals that both compounds do not exhibit significant differences in the level of ATR kinase in the first six hours of incubation, whereas after 12 h of incubation, compound **12a** exhibits significantly higher protein levels than **13d**. Similar elevated levels of ATR kinase are displayed by two control substances taken in the experiment for comparison: rapamycin and valinomycin, and macrocyclic compounds, which is rather natural in terms of the mode of action of these compounds. The changes induced by rapamycin and resemblance of the effects of **12a** and **13d** on ATR expression may be attributed to the dependence effect of the expression of the mTOR and Raptor pair interacting with the ATR–ATRIP pair [52]. Thus, a significant increase in ATR kinase under the influence of compounds **12a** and **13d** may indicate the accumulation of single-stranded DNA breaks. This statement is supported by low levels of the fluorylated *γ*H2AX histone fraction, since the high level of it reliably indicates the accumulation of double-stranded DNA breaks (Figure 6).

ATR kinase and its major downstream effector control kinase 1 (Chk1) prevent cells with damaged or incompletely replicated DNA from entering mitosis when cells are exposed to DNA damaging agents such as ionizing radiation or chemotherapy drugs. ATR activation requires damage to single-stranded DNA (in the G1 period) or replication fork, while ATM kinase responds to double-stranded breaks [52]. Serine/threonine kinase Chk1 is a key downstream regulator of ATR kinase and is phosphorylated at Ser-317 and Ser-345 with its help [53]. Activated Chk1 triggers checkpoints within S and G2/M phases [54]. Most cancer cells have unregulated G1 checkpoints, making them dependent on the S and G2 checkpoints, which are activated through ATR/CHK1 signaling pathways.

Chk1 and Chk2 proteins are serine/threonine kinases with different structure and functions, activated in response to various genotoxic effects [55,56]. Chk2 kinase is a stable protein expressed during all phases of the cell cycle but remains inactive until DNA double-strand breaks occur, and its activation includes dimerization and autophosphorylation. Unlike Chk2, Chk1 kinase is synthesized to a limited extent in the S and G2 phases and is activated in response to DNA damage or replication arrest without autophosphorylation [57]. The main function of Chk1 and Chk2 kinases is checkpoint signaling from the phosphatidylinositol-3-kinase family, especially ATM, ATR and ATX [58], which phosphorylate and activate Chk1 and/or Chk2. ATM and ATR together with Chk2 and Chk1 are involved in the activation and stabilization of the p53 tumor suppressor. In summary, Chk1 and Chk2 kinases regulate fundamental cellular functions such as DNA replication, cell cycle kinetics, chromatin restructuring and apoptosis.

The diagrams in Figure 5 demonstrate that under the influence of compounds **12a** and **13d** in Jurkat tumor cells, the content of phosphorylated checkpoint kinases Chk1 and Chk2 significantly decreases, as when treating with staurosporine, valinomycin, rapamycin and CCCP. Moreover, the decrease in Chk1 and Chk2 is most pronounced in the first six hours after cell treatment. Based on a pronounced decrease in the content of phosphorylated Chk1 and Chk2 kinases, we can assume with great probability that compounds **12a** and **13d** inhibit the cell cycle in both S and G2/M phases, confirmed by cell cycle analysis in Figure 3. Moreover, it is very important for the compounds to affect intracellular signaling, reducing both Chk1 and Chk2 almost simultaneously, thereby effectively preventing cell division, which is of great importance in the development of these kinases as promising molecular targets for antitumor therapy.

Phosphorylation of histone H2AX at serine 139 (*γ*H2AX) is an early stage in the cellular response to DNA damage, when the damage brings about formation of a double-strand break (DSB) [59]. *γ*H2AX foci, considered as markers of DNA double-strand breaks, are easily detected by immunofluorescence microscopy and flow cytometry [60]. The two main signaling pathways ATM/Chk2 and ATR/Chk1 respond to DNA damage and both lead to H2AX phosphorylation, but the ATR/Chk1 pathway causes *γ*H2AX formation to a much lesser extent. The role of *γ*H2AX in DNA repair is also discussed [61,62,63]. The level of *γ*H2AX in the Jurkat cells treated with compounds **12a** and **13d** after 6 and 12 h of incubation was analyzed (Figure 7).

Figure 6 shows that there is no statistically considerable difference between compounds **12a** and **13d** in terms of the effect on the level of phosphorylated *γ*H2AX in Jurkat cells, while staurosporine and rapamycin cause a pronounced increase in the phosphorylated *γ*H2AX histone fraction by almost 9000 times in the first six hours of incubation. These data confirm the previously obtained results of the study of ionic compounds in terms of their ability to activate the internal mitochondrial pathway of apoptosis initiation through the disruption of the processes of oxidation and phosphorylation in mitochondria [17]. What can be stated with a high degree of probability is that pentacyclic triterpenoids-based ionic compounds, with sufficiently effective cell cycle suppression, do not cause pronounced damage to the DNA structure, since the accumulation of *γ*H2AX is minimal in both 6 and 12 h experiments.

The p21 protein (CDKN2A) is one of the key cyclin-dependent kinase inhibitors and an important universal cell cycle protein [64]. It is controlled both by p53 and p53-independent stimuli. Overall, p21 is an important regulator of cell cycle checkpoints to ensure proper cell division. It inhibits cell proliferation directly through binding to Cdks [65] and proliferating cell nuclear antigen (PCNA) [66] or at the transcriptional level [67]. In addition to p53 [68], a large number of oncogenes, tumor suppressors, inflammatory cytokines and nutrients can initiate p21 transcription [69,70]. Its expression increases in response to various intra- and extracellular stimuli to stop the cell cycle, ensuring genome stability. In addition to its role in cell cycle regulation, including mitosis, p21 is involved in differentiation, cell migration, cytoskeletal dynamics, apoptosis, transcription, DNA repair, reprogramming of induced pluripotent stem cells, autophagy and the onset of aging [64]. p21 acts as either a tumor suppressor or an oncogene, depending largely on the cellular context, its subcellular localization and post-translational modifications. Recently, it has become apparent that p53-dependent repression is controlled by the p53–p21–DREAM–E2F/CHR pathway (p53–DREAM pathway) [67]. The p53–DREAM pathway controls over 250 genes, mostly related to the cell cycle. The functional spectrum of these target pathways extends from the G1 phase to the end of mitosis. Therefore, by suppressing the expression of gene products necessary for the passage of the cell cycle, the p53-DREAM pathway is involved in the regulation of all checkpoints from DNA synthesis to cytokinesis, including G1/S, G2/M and the spindle assembly checkpoint. Thus, defects in the p53-DREAM pathway contribute significantly to the loss of regulation of cell cycle checkpoints. Furthermore, the disruption of transcription of p53-DREAM target genes contributes to chromosomal instability and an increase in aneuploidy of cancer cells. Another feature of this pathway is repressing many genes involved in DNA repair and maintenance of telomere length, including contribution to the pathogenesis of Fanconi anemia as well. It is noteworthy that when the DREAM function is lost, cyclin-dependent kinase inhibitor drugs used in cancer treatment, such as Palbociclib, Abemaciclib and Ribociclib, can compensate for defects of this signaling pathway in early stages of cyclin/CDK complexes. Thus, the p53–p21–DREAM–E2F/CHR pathway controls many cell cycle genes, can contribute to cell cycle arrest and is a promising target for cancer treatment. The p53 kinase is able to trans-activate many target genes, with the p21/Waf1 (CDKN2A) gene playing a special role, since its protein product inhibits cyclin-dependent kinase complexes and mediates cell cycle arrest at the G1/S phase boundary under the action of genotoxic stress.

The expression of p21 and p53 proteins was significantly increased in the samples **13d** and **12a** (Figure 8). The concentration of these proteins in both samples increased precisely in the first 6 h of incubation, probably indicating an arrest of the cell cycle in the M2/G phase or attempts of the cell to start a reparation processes. The role of p53 in the Jurkat cell culture is rather difficult to assess, since this cell line expresses the mutant form of this protein, which is unable to start the processes of genome repair or initiate programmed cell death, depending on the level of DNA damage [71,72].

## 4. Conclusions

Thus, oleanolic acid-based ionic compounds proved to be the most effective inducers of apoptosis, which can be explained by a higher solubility and, as a result, better permeability through the biological membranes of these derivatives. It is the ionic form of these compounds that is supposed to have both hydro- and lipophilicity, while the presence of various cations and anions helps to achieve various effects inaccessible to many hybrid molecules when two pharmacophores are linked by a strong covalent bond. This assertion is true for a more precise targeted delivery of the compound and a more complete binding of the ionized pharmacophore to the active center of the target molecule. The mitochondrial pathway of apoptosis in these compounds has been reliably proven by the loss of cytochrome *c* in mitochondria. The initiation of apoptosis by a p53-dependent mechanism can take place as well. Still, it is not the main apoptotic pathway of cells when interacting with ionic compounds based on pentacyclic triterpenoids, since there is no pronounced DNA damage or double-strand breaks accumulation and the level of histones *γ*H2AX is minimal compared to the samples treated with staurosporine, valinomycin and CCCP. The increase in the hypodiploid cell population in the samples with compounds **12a** and **13d**, as well as the arrest of the cell cycle in the G2 and S phases, indicates a static effect of these compounds on the growth and proliferation of the tumor culture, but most likely due to apoptosis initiated by the effect on mitochondria. The activation of STAT5 and STAT3 in the cells treated with the studied ICs is likely to be a constitutive feature of the Jurkat cell culture, or the compounds under study may have a mitogenic effect. The novel ionic compounds synthesized by our group will be further used for complex therapy of hemoblastoses. The used approaches will provide the modifications of many known antitumor drugs that have certain drawbacks due to poor solubility, such as camptothecin or etoposide, by converting them into ionic compounds.

## Data Availability

The data presented in this study are available in this article.

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
