# Peer review of "Pentacyclic Triterpenoids-Based Ionic Compounds: Synthesis, Study of Structure–Antitumor Activity Relationship, Effects on Mitochondria and Activation of Signaling Pathways of Proliferation, Genome Reparation and Early Apoptosis"

_cancers, 2023, doi:10.3390/cancers15030756_

Round 1
Reviewer 1 Report
This manuscript described the synthesis of novel ionic compounds based on three types of pentacyclic triterpene acids (betulinic, oleanolic and ursolic) with these acids acting both as anions and connected through a spacer with various nitrogen-containing compounds and acting as a cation. More than 37 ionic compounds with various counterions were synthesized and their cytotoxicity on the example of various tumor (Jurkat, K562, U937, HL60, A2780) and IC50 values were determined, and the influence of the structure and nature of the anion and cation on the antitumor activity has been specified. By applying modern methods of multiparametric enzyme immunoassay and flow cytometry, intracellular signaling, apoptosis induction, and effects of two active ionic compounds 12a and 13d on the cell cycle and mitochondria have been discussed.
Some approaches and strategies used in this manuscript would provide modification of many known antitumor drugs that have certain drawbacks due to poor solubility.
But the authors should clarify the scientific basis of selecting the tumor cell lines in the manuscript, and should discuss the structure-activity relationships of these pentacyclic triterpene ionic compounds. In particular, the characteristics of the activity and structure of the two active compounds 12a and 13d should be analyzed.
Author Response
Dear Reviewer,
We sent to you revised manuscript "Pentacyclic Triterpenoid’s Based Ionic Compounds: Synthesis, Study of Structure-Antitumor Activity Relationship, Effects on Mitochondria and Activation of Signaling Pathways of Prolif-eration, Genome Reparation and Early Apoptosis" by Lilya U. Dzhemileva, Regina A. Tuktarova, Usein M. Dzhemilev, and Vladimir A. D’yakonov for the publication in the Cancers. We are thanks reviewer for kind support and assistance while editing our paper. We have considered all reviewer's corrections and suggestions. We carefully read the comments of the reviewers and answered to all the comments of the reviewers.
We hope that the readers of the authoritative journal Cancers will find this paper useful for their research, so we ask esteemed Referee to please support our work and accept this article for publication.
Sincerely yours, Lilya Dzhemileva and Vladimir D'yakonov
Point 1: But the authors should clarify the scientific basis of selecting the tumor cell lines in the manuscript, and should discuss the structure-activity relationships of these pentacyclic triterpene ionic compounds. In particular, the characteristics of the activity and structure of the two active compounds 12a and 13d should be analyzed..
Response 1: Thank you for your comment. In this work, we chose the following tumor lines for study - Jurkat, K562, U937, HL60, A2780 and conditionally normal HEK293. We have focused on suspension cultures, because one of the most relevant areas in the development of drugs with antitumor activity is the search for compounds with antileukemic activity. The second aspect was that suspension cultures are more convenient in studying the effects in cells using flow cytometry. And the third aspect is that Jurkat and K562 are p53-deficient lines, and their inclusion in the analysis provides a unique chance to find agents that induce apoptosis in cells bypassing the p53-dependent initiation mechanism. The ovarian cancer line (A2780) was included in the study, since ovarian cancer is also one of the most difficult cancers in terms of localization and the range of drugs for the treatment of ovarian cancer is not very wide.
We conducted structure-activity studies somewhat more widely and unconventionally, since it is very difficult to select compounds with antitumor activity based on cytotoxicity data alone. We approached the solution of this problem in more detail, showing that almost all ionic compounds from our work have a similar mechanism of action, however, the two leaders 12a and 13d have been studied more thoroughly, and a preliminary mechanism of their action in the cell has been shown.
We also discussed our work in sufficient detail for a more accurate and deep understanding of the processes that occur in cells and which proteins are predominantly activated under the action of ionic compounds from triterpenoids.
Once again we heartily thank you for your kind support and assistance while editing our papers. We hope that the readers of the authoritative journal Cancers will find this paper useful for their research, so we ask esteemed Referee to please support our priority and accept this article for publication.
Yours faithfully, Vladimir D'yakonov and Lilya Dzhemileva
Reviewer 2 Report
The authors have done an extensive work which must be appreciated. I think the paper is suitable for publication after major English review and the content need to be concentrated.
The introduction, the discustion, and conclustion need to be extensive revise focus on the topic.
The flow of study need to revised more concentrating.
Author Response
Dear Reviewer,
We sent to you revised manuscript "Pentacyclic Triterpenoid’s Based Ionic Compounds: Synthesis, Study of Structure-Antitumor Activity Relationship, Effects on Mitochondria and Activation of Signaling Pathways of Prolif-eration, Genome Reparation and Early Apoptosis" by Lilya U. Dzhemileva, Regina A. Tuktarova, Usein M. Dzhemilev, and Vladimir A. D’yakonov for the publication in the Cancers. We are thanks reviewer for kind support and assistance while editing our paper. We have considered all reviewer's corrections and suggestions. We carefully read the comments of the reviewers and answered to all the comments of the reviewers.
The article was stylistically revised by an English-speaking researcher in the field of molecular biology and bioorganic chemistry.
We hope that the readers of the authoritative journal Cancers will find this paper useful for their research, so we ask esteemed Referee to please support our work and accept this article for publication.
Sincerely yours, Lilya Dzhemileva and Vladimir D'yakonov
Round 2
Reviewer 2 Report
This paper can be accepted as it is.